# Leaf vein patterning is regulated by the aperture of plasmodesmata intercellular channels

**Nguyen Manh Linh**, **Enrico Scarpella** *

Department of Biological Sciences, University of Alberta, Edmonton, Canada

* enrico.scarpella@ualberta.ca

## Abstract

To form tissue networks, animal cells migrate and interact through proteins protruding from their plasma membranes. Plant cells can do neither, yet plants form vein networks. How plants do so is unclear, but veins are thought to form by the coordinated action of the polar transport and signal transduction of the plant hormone auxin. However, plants inhibited in both pathways still form veins. Patterning of vascular cells into veins is instead prevented in mutants lacking the function of the *GNOM* (*GN*) regulator of auxin transport and signaling, suggesting the existence of at least one more *GN*-dependent vein-patterning pathway. Here we show that in Arabidopsis such a pathway depends on the movement of auxin or an auxin-dependent signal through plasmodesmata (PDs) intercellular channels. PD permeability is high where veins are forming, lowers between veins and nonvascular tissues, but remains high between vein cells. Impaired ability to regulate PD aperture leads to defects in auxin transport and signaling, ultimately leading to vein patterning defects that are enhanced by inhibition of auxin transport or signaling. *GN* controls PD aperture regulation, and simultaneous inhibition of auxin signaling, auxin transport, and regulated PD aperture phenocopies null *gn* mutants. Therefore, veins are patterned by the coordinated action of three *GN*-dependent pathways: auxin signaling, polar auxin transport, and movement of auxin or an auxin-dependent signal through PDs. Such a mechanism of tissue network formation is unprecedented in multicellular organisms.

## Introduction

Most multicellular organisms solve the problem of long-distance transport of water, nutrients, and signaling molecules by tissue networks such as the vascular system of vertebrate embryos and the vein network of plant leaves. How tissue networks form is therefore a key question in developmental biology. In vertebrates, for example, formation of the embryonic vascular system involves direct cell–cell interaction and cell migration (reviewed, for example, in [1,2]). Both those processes are precluded in plants by a cell wall that keeps plant cells apart and in place. Therefore, leaf veins and their networks form by a different mechanism.

How leaf veins form is poorly understood, but the cell-to-cell, polar transport of the plant signaling molecule auxin has long been thought to be both necessary and sufficient for vein

**Data Availability Statement:** All relevant data are within the paper and its Supporting Information files.

**Funding:** This work was supported by a Discovery Grant (Grant Number: RGPIN-2016-04736) of the

Natural Sciences and Engineering Research
Council of Canada (https://www.nserc-crsng.gc.ca)
to ES. NML was supported, in part, by a Summer
Undergraduate Research Fellowship from the
American Society of Plant Biologists. The funders
had no role in study design, data collection and
analysis, decision to publish, or preparation of the
manuscript.

**Competing interests:** The authors have declared
that no competing interests exist.

**Abbreviations:** cals3-2d, callose synthase - 2
dominant; cals3-3d, callose synthase - 3 dominant;
DAG, days after germination; DR5rev, direct repeat
5 reverse; erGFP, endoplasmic-reticulum-localized
GFP; ET, enhancer trap; GN, GNOM; gn-13, gnom -
13; gsl8-1, glucan-synthase-like - 1; gsl8-2,
glucan-synthase-like - 2; gsl8-6, glucan-synthase-
like - 6; gsl8-chor, glucan-synthase-like - chorus;
gsl8-et2, glucan-synthase-like - enlarged tetrad 2;
IAA, indole-3-acetic acid; NPA, N-1-
naphthylphthalamic acid; nYFP, nuclear YFP; PBA,
phenylboronic acid; PD, plasmodesma; PIN1, PIN-
FORMED 1; WT, wild-type.

formation (recently reviewed in [3,4]). Inconsistent with that notion, however, veins still form
in mutants lacking the function of PIN-FORMED (PIN) auxin exporters, whose polar localiza-
tion at the plasma membrane determines the polarity of auxin transport [5–8]. By contrast,
patterning of vascular cells into veins is prevented in mutants lacking the function of the gua-
nine exchange factor for ADP ribosylation factors GNOM (GN): The vascular system of null
gn mutants is no more than a shapeless cluster of randomly oriented vascular cells [5,9–12].

For over 20 years, the vesicle trafficking regulator GN has been thought to perform its
essential vein-patterning function solely through its ability to control the polarity of PIN pro-
tein localization (recently reviewed in [3]). However, two pieces of evidence argue against that
notion. First, the vein patterning defects of gn mutants are quantitatively stronger than and
qualitatively different from those of pin mutants [5]. Second, pin mutations are inconsequen-
tial to the gn vascular phenotype [5]. These observations suggest that other pathways besides
polar auxin transport are involved in vein patterning and that GN controls such additional
pathways too. Such pathways seem to rely on auxin-transporter-independent movement of
auxin or an auxin-dependent signal because pin mutant leaves respond to auxin application by
forming veins that extend away from the auxin application site [5]. Because vein patterning
defects of auxin-transport-inhibited leaves are enhanced by auxin signaling inhibition, the
auxin-transporter-independent movement of auxin or an auxin-dependent signal with vein
patterning function seems to rely, at least in part, on auxin signal transduction [5,13,14]. How-
ever, mutants impaired in both auxin signaling and polar auxin transport only phenocopy
intermediate gn mutants [5], suggesting that additional GN-dependent pathways are involved
in vein patterning.

Because experimental evidence suggests that auxin can move through plasmodesmata
(PDs) intercellular channels (recently reviewed in [15,16]), here we ask whether movement of
auxin or an auxin-dependent signal through PDs is one of the missing GN-dependent vein-
patterning pathways. We find veins are patterned by the coordinated action of three GN-
dependent pathways: auxin signaling, polar auxin transport, and movement of auxin or an
auxin-dependent signal through PDs.

## Results

Available evidence suggests that auxin or an auxin-dependent signal (hereafter collectively
referred to as "auxin signal") moves during leaf development, that such movement is not medi-
ated by known auxin transporters, and that such auxin-transporter-independent movement
controls vein patterning [5]. Here we tested the hypothesis that the movement of an auxin sig-
nal that controls vein patterning and that is not mediated by auxin transporters is enabled by
PDs.

### Control of vein patterning by regulated PD aperture

Should the movement of an auxin signal that controls vein patterning be enabled by PDs,
defects in PD aperture regulation would lead to vein pattern defects. Because severe defects in
the ability to regulate PD aperture lead to embryo lethality (for example, [17–22]), to test the
prediction that defects in PD aperture regulation will lead to vein pattern defects, we analyzed
vein patterns in mature first leaves of the *callose synthase 3 - dominant* (*cals3-d*) and *glucan-
synthase-like 8 / chorus / enlarged tetrad 2 / massue / ectopic expression of seed storage proteins
8* (*gsl8* hereafter) mutants of Arabidopsis, which have, respectively, near-constitutively narrow
and near-constitutively wide PD aperture and can survive embryogenesis [23–29].

Wild-type (WT) Arabidopsis forms broad leaves whose vein networks are defined by at
least four reproducible features: (i) a narrow I-shaped midvein that runs the length of the leaf;

(ii) lateral veins that branch from the midvein and join distal veins to form closed loops; (iii) minor veins that branch from midvein and loops and either end freely or join other veins; and (iv) minor veins and loops that curve near the leaf margin and give the vein network a scalloped outline [5,30–38] (Fig 1A, 1B and 1F). Within individual veins, vascular elements are connected end to end and are aligned along the length of the vein, and free vein ends are as narrow as the rest of the vein (Fig 1G).

*cals3-d* mutants formed narrow leaves whose vein networks deviated from those of WT in at least four respects: (i) fewer veins were formed; (ii) closed loops were often replaced by open loops, i.e., loops that contact the midvein or other loops at only one of their two ends; (iii)

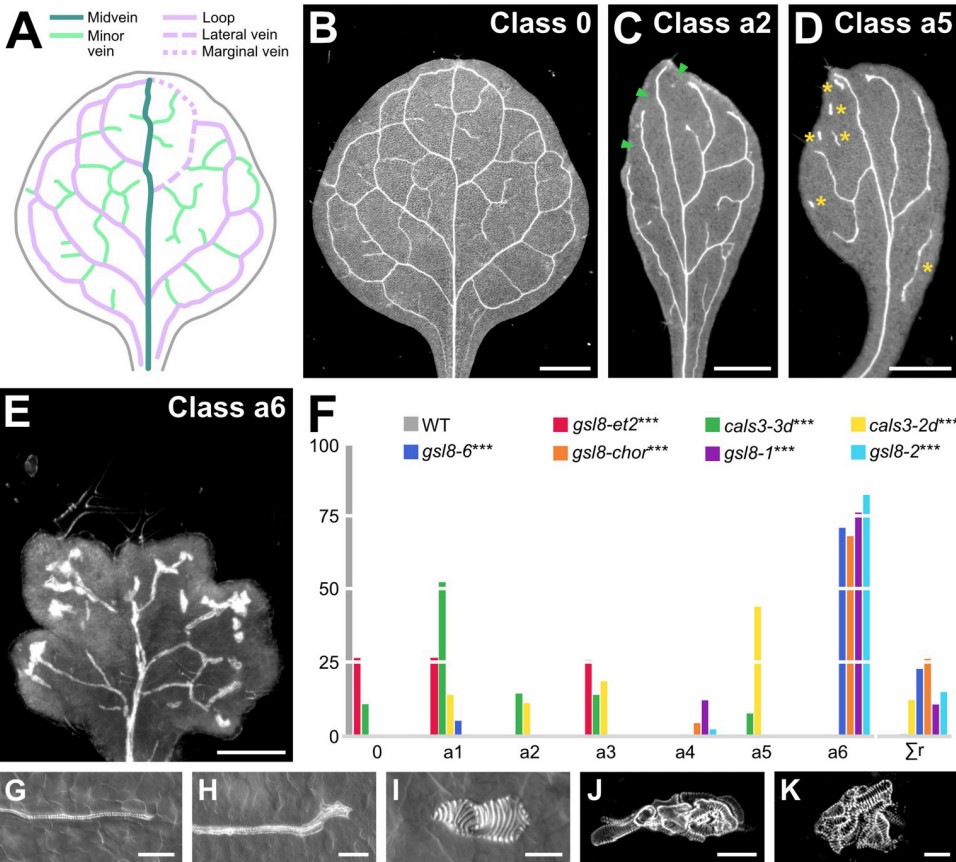

Fig 1. **Control of vein patterning by PD aperture.** (A, B) Vein pattern of mature first leaf of WT Arabidopsis. In (A), teal, midvein; lavender, loops; mint, minor veins. Each loop is formed by the combination of a lateral vein (dashed line) and a marginal vein (dotted line) (only shown, for simplicity, for the first loop on the right side of the leaf). (B–E) Dark-field illumination of mature first leaves illustrating phenotype classes (top right). Class 0: narrow I-shaped midvein and scalloped vein network outline (B); class a2: narrow leaf and open vein network outline (C); class a5: narrow leaf, open vein network outline, and vein fragments and/or vascular clusters (D); class a6: lobed leaf, open vein network outline, and vein fragments and/or vascular clusters (E). Arrowheads: open loops; asterisks: vein fragments and vascular clusters. (F) Percentages of leaves in phenotype classes. Class a1: open vein network outline (S1A Fig); class a3: vein fragments and/or vascular clusters (S1B Fig); class a4: lobed leaf and vein fragments and/or vascular clusters (S1C Fig). Rare vein pattern defects were grouped in class Σr. Difference between *gsl8-et2* and WT, between *cals3-3d* and WT, between *cals3-2d* and WT, between *gsl8-6* and WT, between *gsl8-chor* and WT, between *gsl8-1* and WT, and between *gsl8-2* and WT was significant at *P* < 0.001 (***) by Kruskal-Wallis and Mann-Whitney test with Bonferroni correction. Sample population sizes: WT, 30; *gsl8-et2*, 108; *cals3-3d*, 215; *cals3-2d*, 173; *gsl8-6*, 39; *gsl8-chor*, 45; *gsl8-1*, 65; *gsl8-2*, 47. See also S1 Data. (G–K) Details of veins and vein ends in WT (G) and *cals3-2d* (H) or of vascular clusters in *cals3-2d* (I) and *gsl8-2* (J, K). Differential interference contrast (G–I) or confocal laser scanning (J, K) microscopy. See also S1 Table. Bars: (B–D) 1 mm; (E) 0.25 mm; (G, H, J) 50 μm; (I, K) 25 μm. *cals3-2d, callose synthase - 2 dominant; cals3-3d, callose synthase - 3 dominant; gsl8-1, glucan-synthase-like - 1; gsl8-2, glucan-synthase-like - 2; gsl8-6, glucan-synthase-like - 6; gsl8-chor, glucan-synthase-like - chorus; gsl8-et2, glucan-synthase-like - enlarged tetrad 2*; PD, plasmodesma; WT, wild-type.

veins were often replaced by "vein fragments", i.e., stretches of vascular elements that fail to contact other stretches of vascular elements at either of their two ends, or by "vascular clusters", i.e., isolated clusters of varied sizes and shapes, composed of improperly aligned and connected vascular elements; and (iv) free vein ends often terminated in vascular clusters (Figs 1C, 1D, 1F, 1H, 1I and S1).

Like *cals3-d*, mutants in *GSL8* formed networks of fewer veins in which closed loops were often replaced by open loops; veins were often replaced by vein fragments or isolated vascular clusters; and free vein ends often terminated in vascular clusters (Figs 1E, 1F, 1J, 1K and S1). However, the vein fragments of strong *gsl8* mutants such as *gsl8-2* and *gsl8 - chorus* (*gsl8-chor* hereafter), were shorter, and the clusters were rounder and larger than those of *cals3-d* and weak *gsl8* mutants such as *gsl8 - enlarged tetrad 2* (*gsl8-et2* hereafter) (Figs 1E, 1F, 1J, 1K and S1). Finally, the leaves of strong *gsl8* mutants were smaller than those of WT, *cals3-d*, and weak *gsl8* mutants; and in contrast to the entire leaf-margin of WT, *cals3-d*, and weak *gsl8* mutants, the leaf margin of strong *gsl8* mutants was lobed (Figs 1E, 1F and S1).

That defects in PD aperture regulation led to defects in vein formation, vascular element alignment and connection, and vein continuity and connection is consistent with the hypothesis that movement of an auxin signal through PDs controls vein patterning.

## PD permeability changes during leaf development

Because both near-constitutively wide and near-constitutively narrow PD aperture control vein patterning (Figs 1 and S1), we asked how PD permeability changed during leaf development. To address this question, we expressed an untargeted YFP, which diffuses through PDs [39–41], by the *UAS* promoter, which is inactive in plants in the absence of a GAL4 driver (Fig 2G). We activated YFP expression by tissue- and stage-specific GAL4/erGFP enhancer trap (ET) drivers (Fig 2F) and imaged erGFP expression and YFP signals in first leaves of the resulting ET>>erGFP/YFP plants 2.5 to 6 days after germination (DAG).

In ET>>erGFP/YFP plants, expression of a nondiffusible endoplasmic-reticulum-localized GFP (erGFP) [42] reports expression of GAL4 [43–45], which activates expression of the diffusible YFP. Should the aperture of the PDs in the cells in which YFP expression is activated be narrower than the size of YFP, erGFP and YFP would be detected in the same cells. By contrast, should the aperture be wider than the size of YFP, YFP would be detected in additional cells.

The development of Arabidopsis leaves has been described previously [30,32–34,46–51]. Briefly, at 2 DAG the first leaf is recognizable as a cylindrical primordium with a midvein at its center (Fig 2A). By 2.5 DAG the primordium has expanded (Fig 2B), and by 3 DAG the first loops of veins ("first loops") have formed (Fig 2C). By 4 DAG, a lamina and a petiole have become recognizable, and second loops have formed (Fig 2D). By 5 DAG, lateral outgrowths have become recognizable in the lower quarter of the lamina; third loops have formed; and minor vein have formed in the upper three-quarters of the lamina (Fig 2E). Finally, by 6 DAG minor vein formation has spread to the whole lamina.

In 2.5-DAG primordia of E2331>>erGFP/YFP, in which GAL4 expression is activated at early stages of vein development [52], YFP was detected throughout the 2.5-DAG primordium, even though erGFP was only expressed in the midvein (Fig 2H). At 3 DAG, erGFP was only expressed in the midvein and first loops; nevertheless, YFP was still detected throughout the primordium, even though YFP signals were weaker at the primordium tip (Fig 2I). At 4 DAG, erGFP was only expressed in the midvein and first and second loops (Fig 2J). YFP signals were mainly restricted to the veins in the top half of the 4-DAG leaf but were detected throughout the bottom half of the 4-DAG leaf (Fig 2J). At 5 DAG, erGFP was only expressed in the

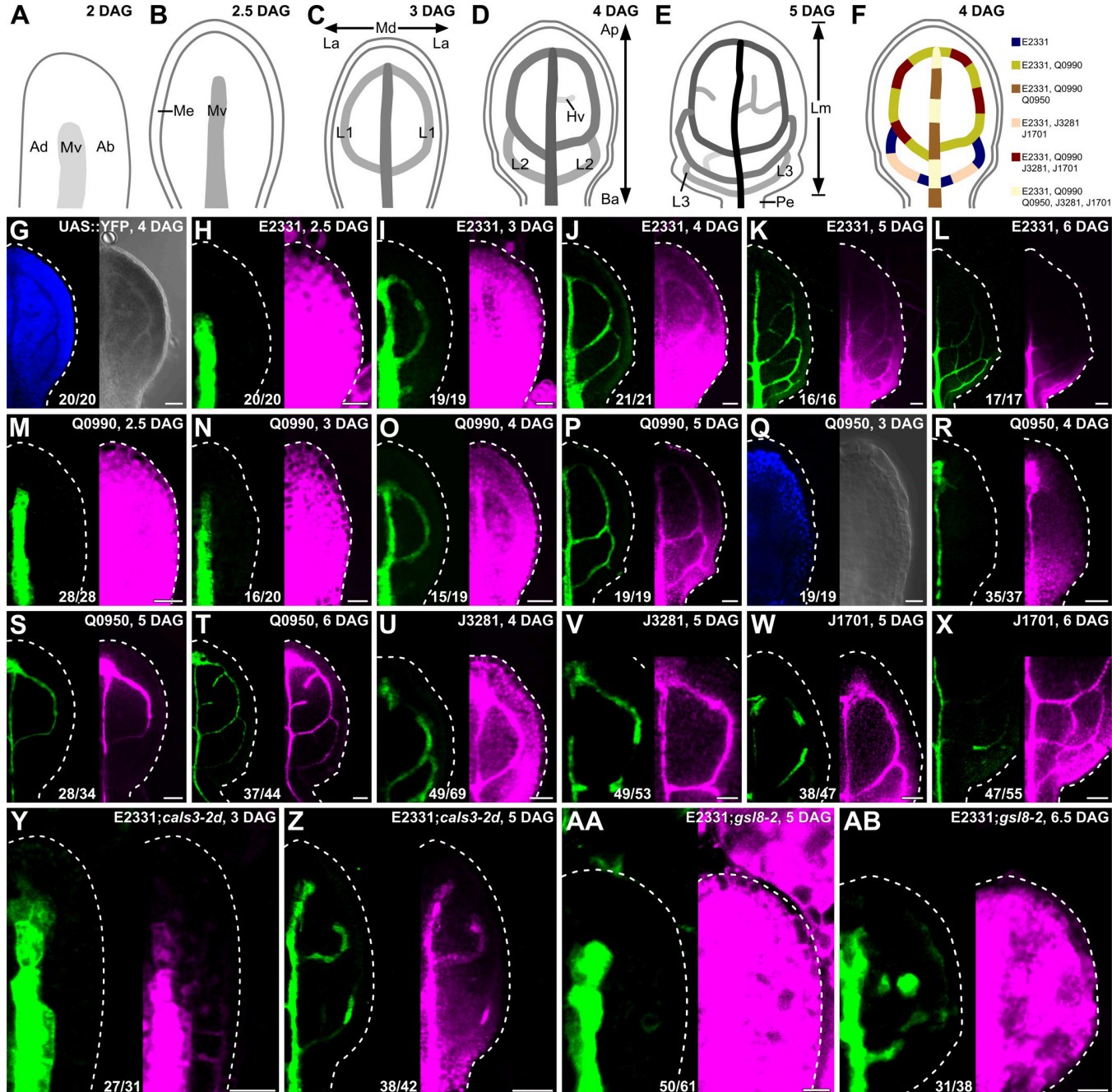

**Fig 2. PD permeability changes during leaf development.** (A–F) Top right: leaf age in DAG. (A–E) Veins form sequentially during Arabidopsis leaf development: The formation of the midvein is followed by the formation of the first loops of veins ("first loops"); the formation of first loops is followed by that of second loops and minor veins; and the formation of second loops and minor veins is followed by that of third loops. Loops and minor veins form in a tip-to-base sequence during leaf development. Increasingly darker grays depict progressively later stages of vein development. Ab, abaxial; Ad, adaxial; Ap, apical; Ba, basal; Hv, minor vein; L1, L2, and L3: first, second, and third loop; La, lateral; Lm, lamina; Md, median; Me, marginal epidermis; Mv, midvein; Pe, petiole. (F) Expression map of tissue- and stage-specific GAL4/erGFP ET drivers in developing leaves illustrates inferred overlap and exclusivity of expression. (G–AB) Differential interference contrast (G, right; Q, right) or confocal laser scanning (all other panels) microscopy. First leaves (for simplicity, only half-leaves are shown). Blue, autofluorescence; green, GFP expression; magenta, YFP signals. Dashed white line delineates leaf outline. Top right: leaf age in DAG and genotype. Bottom center: reproducibility index (see S1 Table). Bars: (H, I, M, N, Q, Y, AA) 20 μm; (G, J, O, R, U) 40 μm; (K, P, S, V, W, Z, AB) 60 μm; (L, T, X) 80 μm. *cals3-2d, callose synthase - 2 dominant*; DAG, days after germination; erGFP, endoplasmic-reticulum-localized GFP; ET, enhancer trap; *gsl8-2, glucan-synthase-like - 2*; PD, plasmodesma.

midvein; first, second, and third loops; and minor veins (Fig 2K). YFP signals were mainly restricted to the veins in the upper three-quarters of the 5-DAG leaf but were detected throughout the lower quarter of the 5-DAG leaf (Fig 2K). At 6 DAG, erGFP continued to be only expressed in the midvein; first, second, and third loops; and minor veins (Fig 2L). YFP signals were mainly restricted to the veins in the whole 6-DAG leaf except for its lowermost part, where YFP was additionally detected in surrounding tissues (Fig 2L).

YFP signals behaved during Q0990>>erGFP/YFP leaf development as they did during E2331>>erGFP/YFP leaf development (Fig 2M–2P; compare with Fig 2H–2L), even though GAL4 expression is activated at intermediate stages of vein development in Q0990, i.e., later than in E2331 [52,53].

In 3-DAG primordia of Q0950>>erGFP/YFP, in which GAL4 is activated at late stages of vein development, i.e., later than in Q0990 [53], neither erGFP nor YFP was expressed (Fig 2Q), further suggesting that the *UAS* promoter is not active in plants in the absence of GAL4. At 4 DAG, erGFP was only expressed in the midvein (Fig 2R). YFP signals were mainly restricted to the midvein in the top half of the 4-DAG leaf but were detected throughout the bottom half of the 4-DAG leaf (Fig 2R). At 5 and 6 DAG, expression of both erGFP and YFP was restricted to the veins (Fig 2S and 2T).

Consistent with previous observations [40], our results suggest that PD permeability is high throughout the leaf at early stages of tissue development. PD permeability remains high in areas of the leaf where veins are still forming, but PD permeability between veins and surrounding nonvascular tissues lowers in areas of the leaf where veins are no longer forming. Eventually, veins become symplastically isolated from surrounding nonvascular tissues. To test whether vein cells become isolated also from one another, we imaged J3281>> and J1701>>erGFP/YFP, in which GAL4 is activated in vein segments [53].

In 4- and 5-DAG J3281>>erGFP/YFP leaves, erGFP was only expressed in segments of midvein and first loops, but YFP was detected in the whole midvein and first loops (Fig 2U and 2V). Likewise, in 5- and 6-DAG J1701>>erGFP/YFP leaves, erGFP was only expressed in segments of midvein and first and second loops, but YFP was detected in whole midvein and loops (Fig 2X and 2W). These results suggest that vein cells are not symplastically isolated from one another even when they are isolated from surrounding nonvascular tissues.

To test whether the reduction in PD permeability between veins and surrounding nonvascular tissues that occurs during normal leaf development were relevant for vein patterning, we imaged E2331>>erGFP/YFP in *cals3-2d* and *gsl8-2* developing leaves.

As in 2.5-DAG E2331>>erGFP/YFP (Fig 2H), in 5-DAG E2331>>erGFP/YFP;*gsl8-2*, erGFP was only expressed in the midvein, and YFP was detected throughout the primordia (Fig 2AA). Also in 3-DAG E2331>>erGFP/YFP;*cals3-2d*, erGFP was only expressed in the midvein (Fig 2Y). However, unlike in E2331>>erGFP/YFP and E2331>>erGFP/YFP;*gsl8-2*, in 3-DAG E2331>>erGFP/YFP;*cals3-2d* YFP signals too were mainly restricted to the midvein—except for its lowermost part, where weak YFP signals were additionally detected in surrounding nonvascular tissues (Fig 2Y).

As in 4-DAG E2331>>erGFP/YFP (Fig 2J), erGFP was only expressed in the vascular tissue of 5-DAG E2331>>erGFP/YFP;*cals3-2d* and 6.5-DAG E2331>>erGFP/YFP;*gsl8-2* (Fig 2Z and 2AB). However, unlike in 4-DAG E2331>>erGFP/YFP (Fig 2J), YFP was mainly restricted to the vascular tissue of the whole 5-DAG E2331>>erGFP/YFP;*cals3-2d* leaf—except for its lowermost part, where weak YFP signals were additionally detected in surrounding nonvascular tissues (Fig 2Z). And YFP signals failed to become restricted to the vascular tissue of 6.5-DAG E2331>>erGFP/YFP;*gsl8-2*, though signal intensity was heterogeneous across the leaf (Fig 2AB).

We conclude that vein patterning defects in *gsl8* and *casl3-d* are, respectively, associated with near-constitutively high and near-constitutively low PD permeability between veins and surrounding nonvascular tissues.

## Auxin-induced vein formation and PD aperture regulation

Auxin application induces vein formation [5,53–57]. Therefore, should the movement of an auxin signal that controls vein patterning be enabled by PDs, defects in PD aperture regulation would lead to defects in auxin-induced vein formation. To test this prediction, we applied the natural auxin indole-3-acetic acid (IAA) to one side of developing first leaves of E2331;*cals3-2d* and Q0990;*gsl8-2* and their respective controls E2331 and Q0990 (Fig 3A, 3B, 3E, and 3F). Because *cals3-2d* and *gsl8* leaves develop more slowly than WT leaves (Fig 4M–4O, 4Q, 4S–4X, 4Z and 4AC), we applied IAA to 3.5-DAG first leaves of E2331 and Q0990 and to 4.5-DAG first leaves of E2331;*cals3-2d* and Q0990;*gsl8-2*. We then assessed erGFP-expression-labeled, IAA-induced vein formation 2.5 days later.

Consistent with previous reports [5,53–57], IAA application induced the formation of veins in approximately 80% (63/79) of E2331 leaves and approximately 90% (22/25) of Q0990 leaves (Fig 3C and 3G). Furthermore, in approximately 55% (34/63) of the E2331 leaves and approximately 65% (14/22) of the Q0990 leaves in which veins formed in response to IAA application, veins readily connected to the midvein (Fig 3C and 3G). By contrast, IAA application induced vein formation in only approximately 60% (27/44) of E2331;*cals3-2d* leaves and approximately 20% (10/56) of Q0990;*gsl8-2* leaves (Fig 3D and 3H). Moreover, only in approximately 30% (8/27) of the E2331;*cals3-2d* leaves in which veins did form in response to IAA application did these veins connect to the midvein (Fig 3D). In the remaining approximately 70% of the responding E2331;*cals3-2d* leaves (19/27), the veins whose formation was induced by IAA application ran parallel to the midvein through the leaf petiole (Fig 3D). Conversely, in 90% (9/10) of the Q0990;*gsl8-2* leaves in which IAA induced vein formation, not only did the veins whose formation was induced by IAA application connect to the midvein, but they did so by expanding into a broad vascular differentiation zone (Fig 3H). Therefore, both near-constitutively wide and near-constitutively narrow PD aperture inhibit auxin-induced vein formation. Whenever the tissue escapes such an effect, near-constitutively narrow PD aperture inhibits connection of newly formed veins to preexisting ones, and near-constitutively wide PD aperture accentuates that connection through excess vascular differentiation. These observations suggest that auxin-induced vein formation depends on regulated PD aperture, that restriction of auxin-induced vascular differentiation to limited cell files depends on narrow PD aperture, and that connection of veins whose formation is induced by auxin depends on wide PD aperture. Were that so, auxin application would impinge on the reduction in PD permeability between veins and surrounding nonvascular tissues that occurs during normal leaf development (Fig 2). To test this prediction, we applied IAA to one side of 3.5-DAG first leaves of E2331>>erGFP/YFP and assessed erGFP-expression-labeled, IAA-induced vein formation and YFP-signal-inferred PD permeability 2.5 and 4.5 days later (i.e., 6 and 8 DAG, respectively).

Consistent with what is shown above (Figs 2I, 2J and 3A), at 3.5 DAG erGFP was only expressed in the midvein and in first and second loops (Fig 3I). In the top half of 3.5-DAG leaves, YFP signals were weaker in nonvascular tissues than in veins but were uniformly strong in the bottom half of the leaves (Fig 3I). As shown above (Fig 2L), at 6 DAG erGFP expression was restricted to the veins (Fig 3J and 3K). In 6-DAG leaves to which IAA had not been applied, YFP signals were mainly restricted to the veins in the whole leaf except for its lowermost part, where YFP was additionally detected in surrounding tissues (Fig 3J). By contrast,

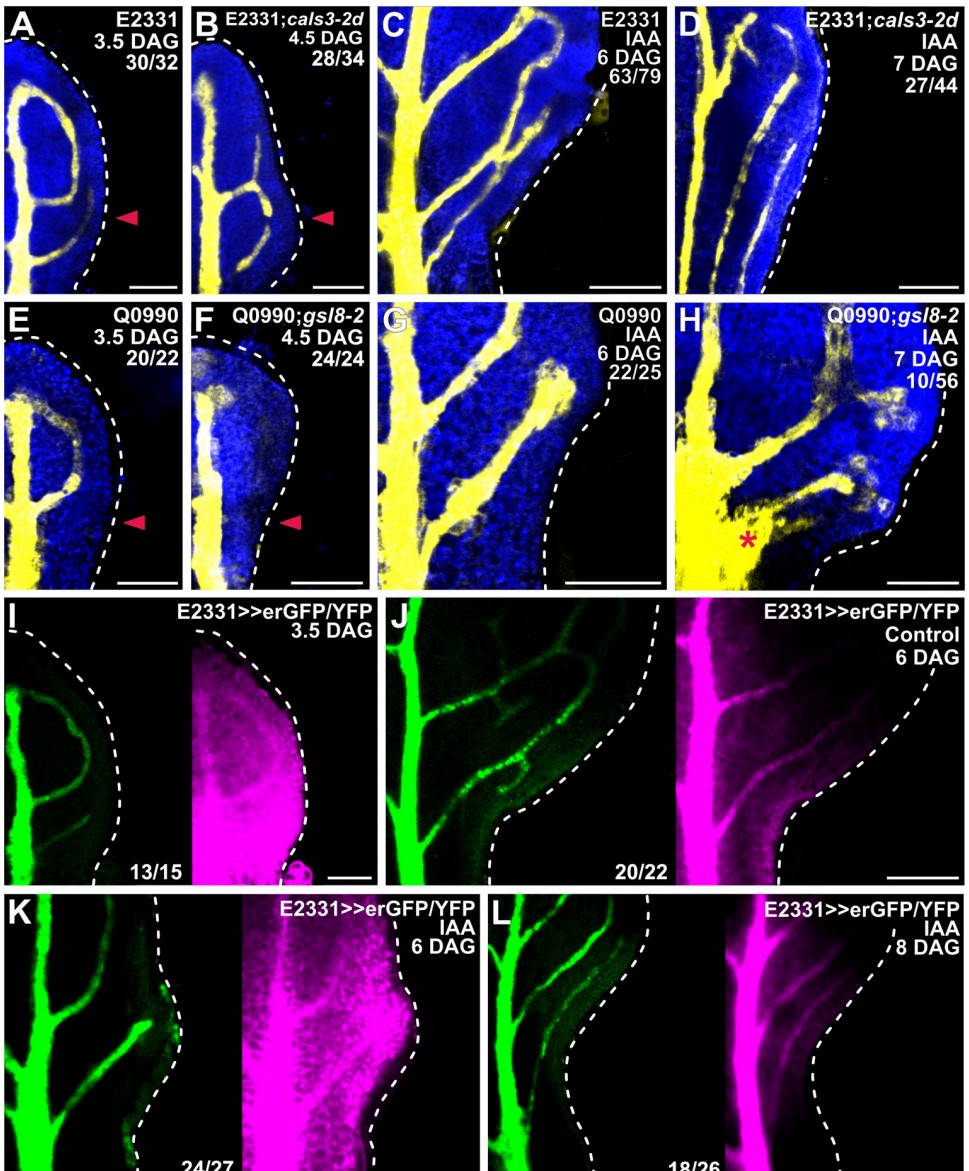

**Fig 3. Auxin-induced vein formation and PD aperture regulation.** (A–L) Confocal laser scanning microscopy. First leaves (for simplicity, only half-leaves are shown). Blue, autofluorescence; yellow (A–H) or green (I–L), GFP expression; magenta, YFP signals. Dashed white line delineates leaf outline. Top right: leaf age in DAG, genotype, treatment, and—in A–H—reproducibility index (see S1 Table). Bottom center (I–L): reproducibility index. Arrowhead in A, B, E, F indicates position of IAA application. Star in H indicates broad area of vascular differentiation connecting the midvein with the vein whose formation was induced by IAA application. Bars: (A, E, F, G–I, K) 80 μm; (B) 60 μm; (C, J, L) 120 μm; (D) 150 μm. *cals3-2d, callose synthase - 2 dominant*; DAG, days after germination; endoplasmic-reticulum-localized GFP; *gsl8-2, glucan-synthase-like - 2*; IAA, indole-3-acetic acid; PD, plasmodesma.

YFP signals were detected throughout the bottom half of the 6-DAG leaves to which IAA had been applied (Fig 3K). By 8 DAG, however, both erGFP and YFP signals had become mainly restricted to the veins also in the leaves to which IAA had been applied (Fig 3L).

In conclusion, our results suggest that auxin application delays the reduction in PD permeability between veins and surrounding nonvascular tissues that occurs during normal leaf development and that auxin-induced vein formation and connection depend on the ability to

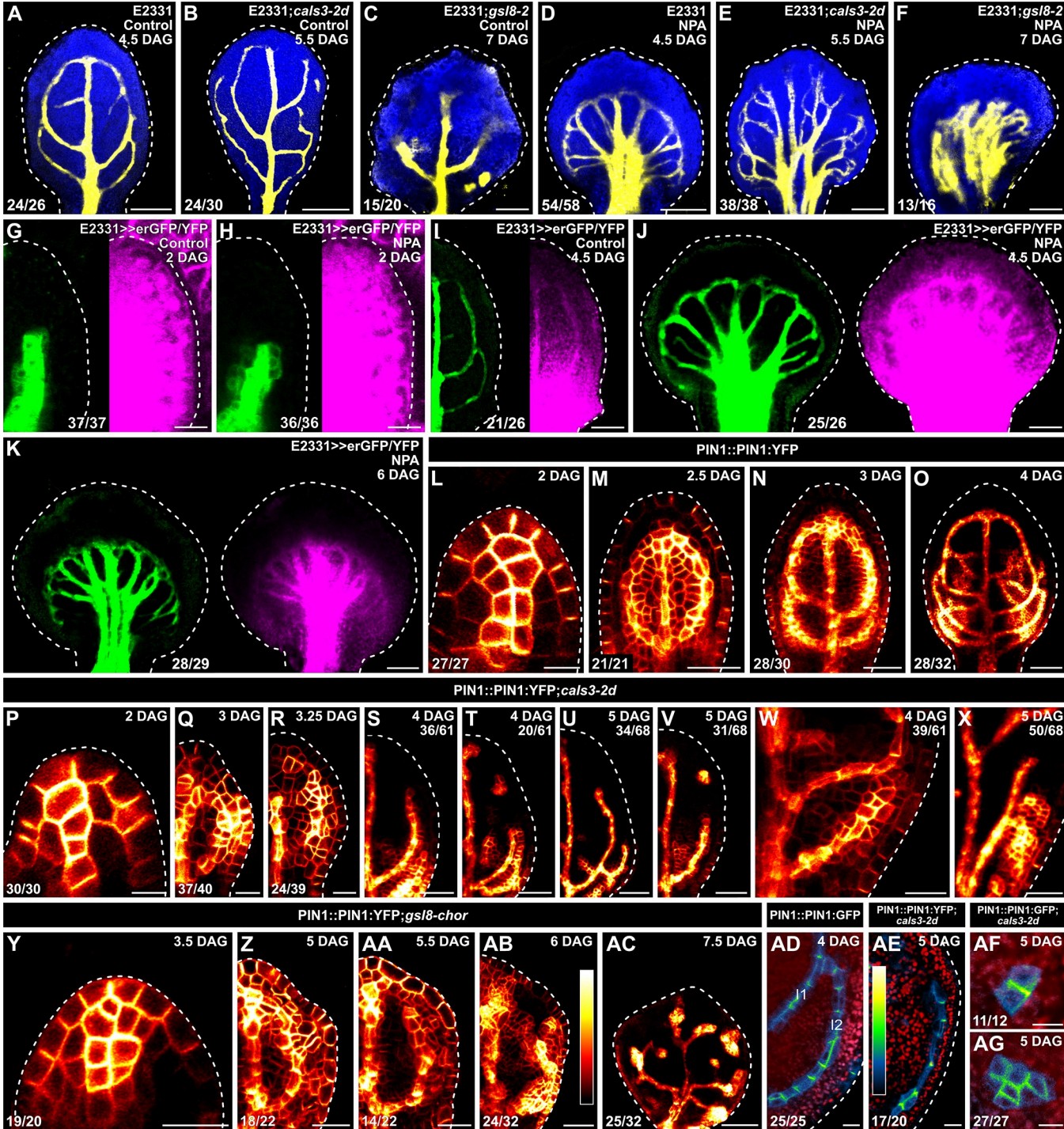

**Fig 4. Auxin-transport-dependent vein patterning and regulated PD aperture.** (A–AG) Confocal laser scanning microscopy. First leaves (for simplicity, only half-leaves are shown in G–I, Q–V, and Z–AB). Blue (A–F) or red (AD–AG), autofluorescence; yellow (A–F) or green (G–K), GFP expression; magenta, YFP signals. Dashed white line delineates leaf outline. (G, H, L, P, Y) Side view, adaxial side to the left. (L–AC) PIN1::PIN1:YFP expression; look-up table (ramp in AB) visualizes expression levels. (AD–AG) PIN1::PIN1:GFP (AD, AF, AG) or PIN1::PIN1:YFP (AE) expression; look-up table (ramp in AE) visualizes expression levels. Top right: leaf age in DAG, genotype, and treatment (25 μM NPA), and—in S–V, X—reproducibility index. Bottom left (A–F, L–R, W, Y–AG) or center (G–K): reproducibility index (see S1 Table). Bars: (A–F, I–K) 120 μm; (G, H, Q, R, W, X, Y) 20 μm; (L, P, AD–AG) 10 μm; (M, N, S, T, Z, AA, AB) 40 μm; (O, U, V, AC) 60 μm. *cals3-2d, callose synthase - 2 dominant*; DAG, days after germination; erGFP, endoplasmic-reticulum-localized GFP; *gsl8-2, glucan-synthase-like - 2*; *gsl8-chor, glucan-synthase-like - chorus*; NPA, N-1-naphthylphthalamic acid; PD, plasmodesma; PIN1, PIN-FORMED 1.

regulate PD aperture. Such conclusions are consistent with the hypothesis that the movement of an auxin signal that controls vein patterning is enabled by PDs.

## Auxin-transport-dependent vein patterning and regulated PD aperture

Should the movement of an auxin signal that controls vein patterning and is not mediated by auxin transporters be enabled by PDs, defects in PD aperture regulation would enhance vein patterning defects induced by auxin transport inhibition. To test this prediction, we grew E2331, E2331;*cals3-2d*, and E2331;*gsl8-2* in the presence or absence of the auxin transport inhibitor N-1-naphthylphthalamic acid (NPA) [58], which binds PIN proteins and inhibits their activity [59,60] and which induces vein patterning defects that phenocopy the loss of that *PIN*-dependent auxin transport pathway that is relevant for vein patterning [5]. We then imaged erGFP-expression-labeled vein networks 4.5 DAG in E2331. Because *cals3-2d* and *gsl8* leaves develop more slowly than WT leaves (Fig 4M–4O, 4Q, 4S–4X, 4Z and 4AC), we imaged erGFP-expression-labeled vein networks 5.5 and 7 DAG in E2331;*cals3-2d* and E2331;*gsl8-2*, respectively.

Consistent with what is shown above (Fig 2J and 2K), in the absence of NPA vein networks were composed of midvein, first and second loops, and minor veins in E2331, and of midvein, loops—whether open or closed—vein fragments, and vascular clusters in E2331;*cals3-2d* and E2331;*gsl8-2* (Fig 4A–4C).

Consistent with previous reports [5,34,35], NPA reproducibly induced characteristic vein pattern defects in E2331 leaves: (i) the vein network comprised more lateral-veins; (ii) lateral veins failed to join the midvein in the middle of the leaf and instead ran parallel to one another to form a wide midvein; and (iii) lateral veins joined distal veins in a marginal vein that closely paralleled the leaf margin and gave a smooth outline to the vein network (Fig 4D).

As in E2331, in E2331;*cals3-2d* NPA induced the formation of more lateral-veins that failed to join the midvein in the middle of the leaf and instead ran parallel to one another to form a wide midvein (Fig 4E). However, unlike in NPA-grown E2331, in NPA-grown E2331;*cals3-2d* lateral veins often failed to join distal veins in a marginal vein and instead ended freely in the lamina near the leaf margin (Fig 4E).

As in both E2331 and E2331;*cals3-2d*, in E2331;*gsl8-2* NPA induced the formation of more veins, but these veins ran parallel to one another to give rise to a midvein that spanned almost the entire width of the leaf (Fig 4F). And as in NPA-grown E2331;*cals3-2d*—but unlike in NPA-grown E2331—in NPA-grown E2331;*gsl8-2* only rarely did veins join one another in a marginal vein; instead, they most often ended freely in the lamina near the leaf tip (Fig 4F).

In conclusion, both near-constitutively wide and near-constitutively narrow PD aperture enhance vein patterning defects induced by auxin transport inhibition, a conclusion that is consistent with the hypothesis that the movement of an auxin signal that controls vein patterning and is not mediated by auxin transporters is enabled by PDs. Moreover, because auxin transport inhibition promotes vein connection [38], that NPA was unable to induce vein connection in E2331;*cals3-2d* and E2331;*gsl8-2* suggests that the promoting effect of auxin transport inhibition on vein connection depends on regulated PD aperture. Were that so, auxin transport inhibition would impinge on the reduction of PD permeability that occurs between veins and surrounding nonvascular tissues during normal leaf development. To test this prediction, we grew E2331>>erGFP/YFP in the presence or absence of NPA and assessed erGFP-expression-labeled vein network formation and YFP-signal-inferred PD aperture in first leaves 2, 4.5, and 6 DAG.

Consistent with what is shown above (Fig 2H), at 2 DAG erGFP was only expressed in the midvein of both NPA- and normally grown E2331>>erGFP/YFP—though the erGFP

expression domain was broader in NPA-grown than in normally grown primordia (Fig 4G and 4H). Likewise, in both NPA- and normally grown E2331>>erGFP/YFP YFP was detected throughout the 2-DAG primordia (Fig 4G and 4H).

Also consistent with what is shown above (Fig 2J and 2K), at 4.5 DAG erGFP was only expressed in the midvein, first and second loops, and minor veins of normally grown E2331>>erGFP/YFP (Fig 4I). YFP signals were mainly restricted to the veins in the top half of normally grown 4.5-DAG E2331>>erGFP/YFP leaves but were detected throughout the bottom half of the leaves (Fig 4I). Also in NPA-grown 4.5-DAG E2331>>erGFP/YFP leaves, erGFP expression was restricted to the veins; however, YFP was detected throughout NPA-grown 4.5-DAG E2331>>erGFP/YFP leaves—though YFP signals were weaker along the margin in the top half of the leaves (Fig 4J). Nevertheless, by 6 DAG both erGFP and YFP signals had become mainly restricted to the veins also in NPA-grown E2331>>erGFP/YFP (Fig 4K).

We conclude that auxin transport inhibition delays the reduction in PD permeability between veins and surrounding nonvascular tissues that occurs during normal leaf development and that such delay mediates the promoting effect of auxin transport inhibition on vein connection.

We next asked whether regulated PD aperture in turn controlled polar auxin transport during leaf development. Because *PIN1* is the only auxin-transporter-encoding gene in Arabidopsis with nonredundant functions in vein patterning [37], to address that question we imaged domains and cellular localization of expression of PIN1::PIN1:YFP (PIN1:YFP fusion protein expressed by the *PIN1* promoter [61]) or PIN1::PIN1:GFP [62] during first-leaf development in WT, *cals3-2d*, and *gsl8-chor*.

Consistent with previous reports [5,37,38,56,62–67], in WT PIN1::PIN1:YFP was expressed in all the cells at early stages of tissue development, and inner tissue expression was stronger in developing veins (Fig 4L–4O). Over time, epidermal expression became restricted to the basal-most cells, and inner tissue expression became restricted to developing veins (Fig 4L–4O).

Also in *cals3-2d* and *gsl8-chor*, PIN1::PIN1:YFP was expressed in all the cells at early stages of tissue development, and inner tissue expression was stronger in developing veins (Fig 4P–4AC). Furthermore, as in WT, in both *cals3-2d* and *gsl8-chor* PIN1::PIN1:YFP expression domains associated with loop formation were initially connected on both ends to preexisting expression domains, and PIN1::PIN1:YFP was evenly expressed along those looped domains (Fig 4Q and 4Z). However, in *casl3-2d* and *gsl8-chor* PIN1::PIN1:YFP expression along looped domains soon became heterogeneous, with domain segments with stronger expression separated by segments with weaker expression (Fig 4R, 4W, 4AA, and 4AB). Such heterogeneity in PIN1::PIN1:YFP expression at early stages of loop formation was associated with open or fragmented looped domains of PIN1::PIN1:YFP expression at later stages (Fig 4S–4V, 4X and 4AC). Finally, equivalent stages of vein development occurred at later time points in *casl3-2d* and *gsl8-chor* than in WT (for example, compare Fig 4Q and 4Z with Fig 4M, Fig 4S, 4T and 4W with Fig 4N, and Fig 4U, 4V, 4X and 4AC with Fig 4O).

Consistent with previous reports [5,37,38,56,65–67], in cells at late stages of second loop development in WT leaves, by which time PIN1::PIN1:GFP expression had become restricted to the cells of the developing loop, PIN1::PIN1:GFP expression was polarly localized to the side of the plasma membrane facing the veins to which the second loop was connected (Fig 4AD).

By contrast, in cells at late stages of development of vein fragments and isolated vascular clusters in *cals3-2d*, by which time expression domains of PIN1::PIN1:GFP or PIN1::PIN1:YFP (hereafter collectively referred to as PIN1::PIN1:FP) had become disconnected from other veins on both ends, PIN1::PIN1:FP expression was polarly localized to any of the plasma membrane sides facing a contiguous PIN1::PIN1:FP-expressing cell (Fig 4AE–4AG).

In conclusion, our results suggest that vein patterning is controlled by the mutually coordinated action of polar auxin transport and movement of an auxin signal through PDs.

## Auxin-signaling-dependent vein patterning and regulated PD aperture

The movement of an auxin signal that controls vein patterning and is not mediated by auxin transporters depends, in part, on auxin signaling [5]. Should the residual, auxin-transporter-and auxin-signaling-independent movement of an auxin signal that controls vein patterning be enabled by PDs, defects in PD aperture regulation would enhance vein patterning defects induced by auxin signaling inhibition—just as defects in PD aperture enhance vein patterning defects induced by auxin transport inhibition (Fig 4A–4F). To test the prediction that defects in PD aperture will enhance vein patterning defects induced by auxin signaling inhibition, we grew E2331, E2331;*cals3-2d*, and E2331;*gsl8-2* in the presence or absence of the auxin signaling inhibitor phenylboronic acid (PBA) [68], which induces vein patterning defects that phenocopy the loss of that *AUXIN RESISTANT 1* -, *TRANSPORT INHIBITOR RESPONSE 1 / AUXIN SIGNALING F-BOX 2* -, and *MONOPTEROS*-dependent auxin signaling pathway that is relevant for vein patterning [5,68]. We then imaged erGFP-expression-labeled vein networks 4.5 DAG in E2331. Because *cals3-2d* and *gsl8* leaves develop more slowly than WT leaves (Fig 4M–4O, 4Q, 4S–4X, 4Z and 4AC), we imaged erGFP-expression-labeled vein networks 5.5 and 6.5 DAG in E2331;*cals3-2d* and E2331;*gsl8-2*, respectively.

As shown above (Fig 4A–4C), in the absence of PBA vein networks were composed of midvein, first and second loops, and minor veins in E2331, and of midvein, loops—whether open or closed—vein fragments, and vascular clusters in E2331;*cals3-2d* and E2331;*gsl8-2* (Fig 5A, 5D and 5G).

Ten μM PBA failed to induce vein network defects in E2331 but led to the formation of fewer veins and opening or fragmentation of all the loops in E2331;*cals3-2d* and E2331;*gsl8-2* (Fig 5B, 5E, 5H and 5I). Formation of fewer veins and opening of all the loops—though not their fragmentation—were induced in E2331 by 50 μM PBA (Fig 5C). At that concentration of PBA, the vascular systems of E2331;*cals3-2d* and E2331;*gsl8-2* were mainly composed of very few, scattered vascular clusters (Fig 6F, 6J and 6K).

These observations suggest that defects in PD aperture enhance vein patterning defects induced by auxin signaling inhibition, a conclusion that is consistent with the hypothesis that the residual, auxin-transporter-and auxin-signaling-independent movement of an auxin signal that controls vein patterning is enabled by PDs. Moreover, these observations suggest that the vein patterning defects induced by auxin signaling inhibition may, at least in part, depend on regulated PD aperture. Were that so, auxin signaling inhibition would impinge on the reduction of PD permeability that occurs between veins and surrounding nonvascular tissues during normal leaf development. To test this prediction, we grew E2331>>erGFP/YFP in the presence or absence of PBA and assessed erGFP-expression-labeled vein network formation and YFP-signal-inferred PD aperture in first leaves 2 and 4.5 DAG.

As shown above (Fig 4G), at 2 DAG erGFP was only expressed in the midvein of both PBA-and normally grown E2331>>erGFP/YFP (Fig 5L and 5M). Likewise, in both PBA- and normally grown E2331>>erGFP/YFP YFP was detected throughout the 2-DAG primordia (Fig 5L and 5M).

As also shown above (Fig 4I), at 4.5 DAG erGFP was only expressed in the midvein, first and second loops, and minor veins of normally grown E2331>>erGFP/YFP (Fig 5N). YFP signals were mainly restricted to the veins in the top half (49.83% ± 0.73, *n* = 20; S2 Data) of normally grown 4.5-DAG E2331>>erGFP/YFP leaves but were detected throughout the bottom half of the leaves (Fig 5N). Also in PBA-grown 4.5-DAG E2331>>erGFP/YFP leaves,

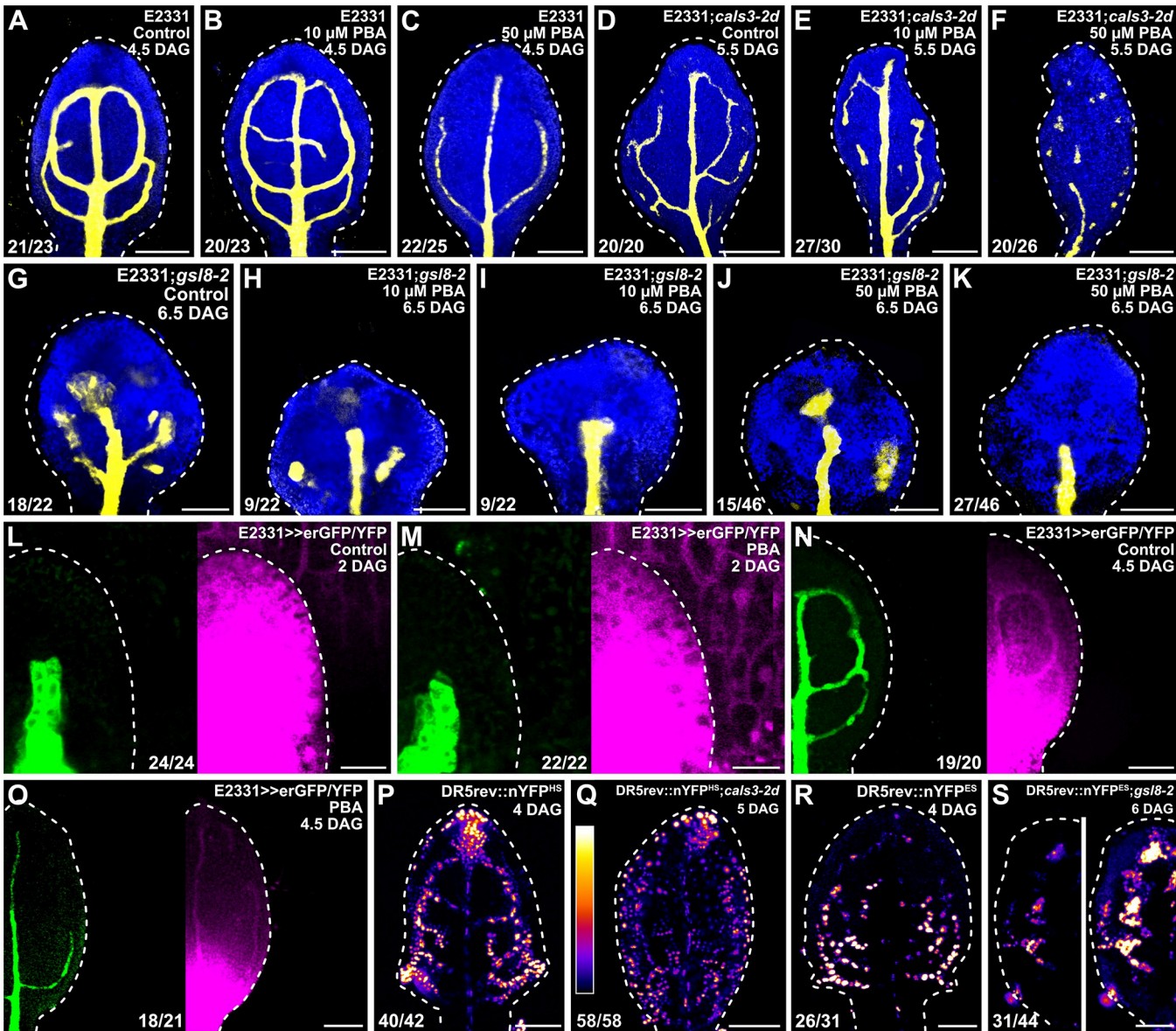

**Fig 5. Auxin-signaling-dependent vein patterning and regulated PD aperture.** (A–S) Confocal laser scanning microscopy. First leaves (for simplicity, only half-leaves are shown in L–O, S). Blue, autofluorescence; yellow (A–K) or green (L–O), GFP expression; magenta, YFP signals. Dashed white line delineates leaf outline. (L, M) Side view, adaxial side to the left. (P–S) DR5rev::nYFP$^{HS}$ (P,Q) or DR5rev::nYFP$^{ES}$ (R,S) expression; look-up table (ramp in Q) visualizes expression levels. Top right: leaf age in DAG, genotype, and treatment (10 or 50 μM PBA). Bottom left (A–K, P–S) or center (L–O): reproducibility index (see S1 Table). Images in P and R were acquired by matching signal intensity to detector's input range (approximately 1% saturated pixels). Images in P and Q were acquired at identical settings and show weaker and broader DR5rev::nYFP$^{HS}$ expression in *cals3-2d*. Images in R and S (left) were acquired at identical settings and show weaker DR5rev::nYFP$^{ES}$ expression in *gsl8-2*. Image in S (right) was acquired by matching signal intensity to detector's input range (approximately 1% saturated pixels) and shows broader DR5rev::nYFP$^{ES}$ expression in *gsl8-2*. Bars: (A–K) 120 μm; (L, M) 20 μm; (N–S) 80 μm. *cals3-2d, callose synthase - 2 dominant*; DAG, days after germination; DR5rev, direct repeat 5 reverse; erGFP, endoplasmic-reticulum-localized GFP; *gsl8-2, glucan-synthase-like - 2*; nYFP, nuclear YFP; PBA, phenylboronic acid; PD, plasmodesma.

erGFP expression was restricted to the veins; however, YFP signals were already restricted to the veins in the top two-thirds (66.78% ± 0.63, *n* = 21, *P* < 0.001; S2 Data) of PBA-grown 4.5-DAG E2331>>erGFP/YFP leaves (Fig 5O), suggesting that auxin signaling inhibition leads to premature reduction in PD permeability.

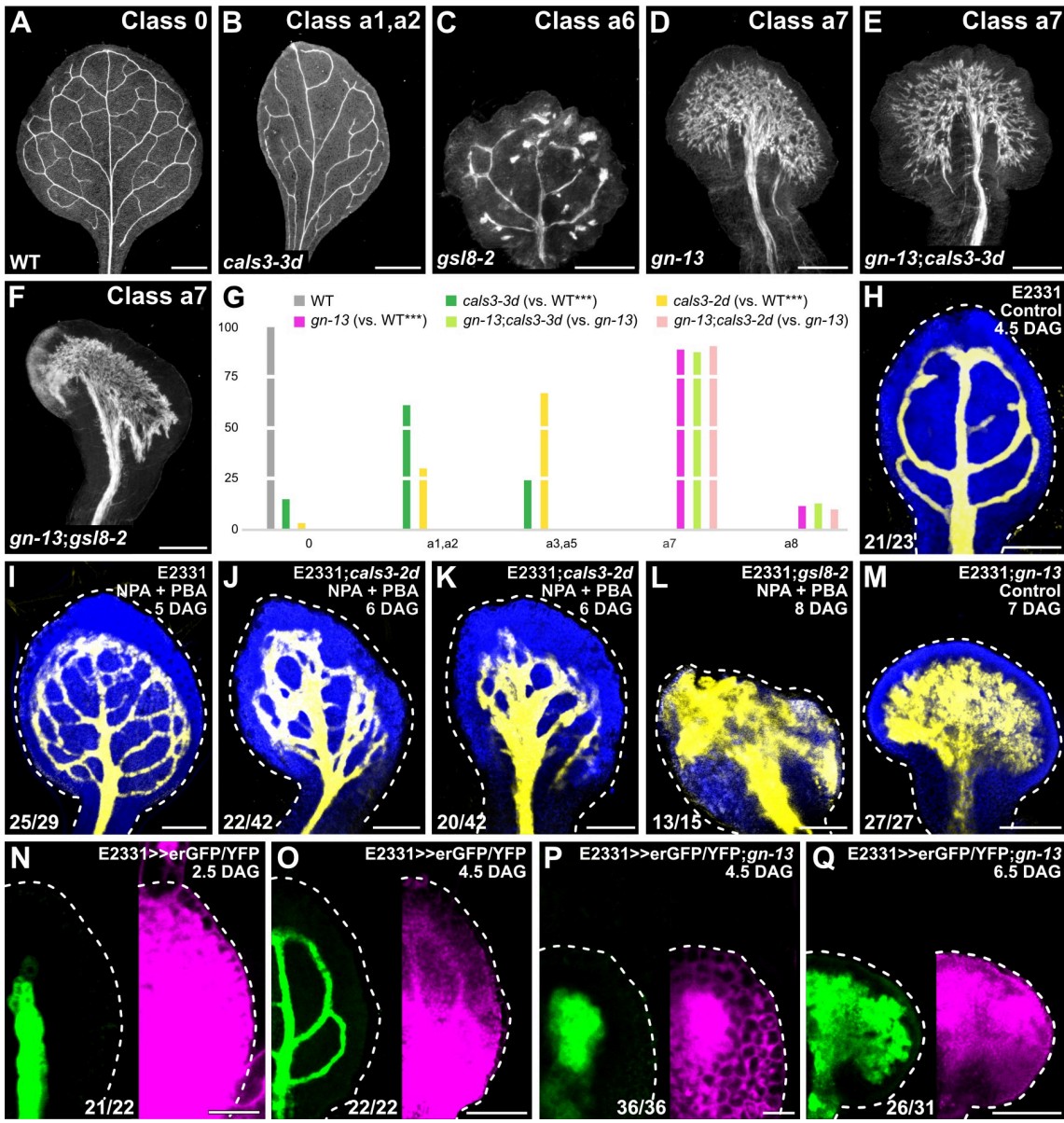

**Fig 6. Control of PD-aperture-dependent vein patterning by *GNOM*.** (A–F) Dark-field illumination of mature first leaves illustrating phenotype classes (top right) and genotypes (bottom left). Classes 0, a1, a2, and a6 defined in Fig 1. Class a7: wide midvein and shapeless vascular cluster (D–F). (G) Percentages of leaves in phenotype classes. Classes 0, a1–a3, a5, and a6 defined in Fig 1. Class a8, shapeless vascular cluster. Difference between *cals3-3d* and WT, between *cals3-2d* and WT, and between *gn-13* and WT was significant at $P < 0.001$ (***) by Kruskal-Wallis and Mann-Whitney test with Bonferroni correction. Sample population sizes: WT, 30; *cals3-3d*, 62; *cals3-2d*, 67; *gn-13*, 89; *gn-13;cals3-3d*, 100; *gn-13;cals3-2d*, 52. See S3 Data. (H–Q) Confocal laser scanning microscopy. First leaves. Blue, autofluorescence; yellow (H–M) or green (N–Q), GFP expression; magenta, YFP signals. Dashed white line delineates leaf outline. Top right: leaf age in DAG, genotype, and treatment (25 μM NPA + 10 μM PBA). Bottom left: reproducibility index (see S1 Table). Bars: (A, B) 1 mm; (C, D, F) 0.5 mm; (E) 0.25 mm; (H, J, K, O) 100 μm; (I, L, M, Q) 150 μm; (N, P) 25 μm. *cals3-2d, callose synthase - 2 dominant*; *cals3-3d, callose synthase - 3 dominant*; DAG, days after germination; erGFP, endoplasmic-reticulum-localized GFP; *gn-13, gnom - 13*; *gsl8-2, glucan-synthase-like - 2*; NPA, N-1-naphthylphthalamic acid; PBA, phenylboronic acid; PD, plasmodesma; WT, wild-type.

We conclude that auxin signaling inhibition prematurely reduces PD permeability between veins and surrounding nonvascular tissues and that such premature reduction mediates, at least in part, the effects of auxin signaling inhibition on vein patterning.

We next asked whether regulated PD aperture in turn controlled response to auxin signals in developing leaves. To address this question, we imaged expression of the auxin response reporter DR5rev::nYFP [37,64] (S2 Data) 4 DAG in WT and—because *cals3-2d* and *gsl8* leaves develop more slowly than WT leaves (Fig 4M–4O, 4Q, 4S–4X, 4Z, and 4AC)—5 and 6 DAG in *cals3-2d* and *gsl8-2*, respectively.

As previously shown [5,37,38], in WT strong DR5rev::nYFP expression was mainly associated with developing veins (Fig 5P and 5R). By contrast, DR5rev::nYFP expression was weaker and expression domains were broader in *cals3-2d* and *gsl8-2* (Fig 5Q and 5S).

In conclusion, our results suggest that vein patterning is controlled by the mutually coordinated action of auxin signaling and movement of an auxin signal through PDs.

## Control of PD-aperture-dependent vein patterning by *GN*

Vein patterning is controlled by the mutually coordinated action of auxin signaling, polar auxin transport, and movement of an auxin signal through PDs (Figs 1–5). Vein patterning activities of both auxin signaling and polar auxin transport depend on *GN* function [5]. We asked whether *GN* also controlled PD-aperture-dependent vein patterning. To address this question, we compared the phenotypes of mature first leaves of the *gn-13*;*cals3-2d*, *gn-13*;*cals3-3d*, and *gn-13*;*gsl8-2* double mutants with those of their respective single mutants.

The phenotypes of *gn-13*;*cals3-2d* and *gn-13*;*cals3-3d* were no different from those of *gn-13* (Fig 6A, 6B, 6D, 6E and 6G), suggesting that the effects of the *gn-13* mutation on vein patterning are epistatic to those of the *cals3-d* mutation. Furthermore, the *gn* phenotype segregated in approximately one-quarter (559/2,353) of the progeny of plants heterozygous for both *gn-13* and *gsl8-2*—no different from the frequency expected by the Pearson's chi-squared ($\chi^2$) goodness of fit test for the hypothesis that the phenotype of *gn-13* is epistatic to that of *gsl8-2*. We confirmed by genotyping that some of the *gn*-looking seedlings are indeed *gn-13*;*gsl8-2* double homozygous mutants whose leaves are no different from those of *gn-13* (Fig 6A, 6C, 6D and 6F), suggesting that the effects of the *gn-13* mutation on vein patterning are epistatic to those of the *gsl8-2* mutation. These observations suggest that *GN* controls PD-aperture-dependent vein patterning; were that so, *gn* leaves would have defects in regulation of PD permeability. To test this prediction, we imaged E2331>>erGFP/YFP in developing *gn-13* leaves.

In both E2331>>erGFP/YFP and E2331>>erGFP/YFP;*gn-13*, erGFP was only expressed in the vascular tissue (Fig 6N–6Q). Furthermore, in both 2.5-DAG E2331>>erGFP/YFP and 4.5-DAG E2331>>erGFP/YFP;*gn-13* YFP was detected throughout the primordium; however, in 4.5-DAG E2331>>erGFP/YFP;*gn-13* YFP signals were weaker in nonvascular tissues than in the vascular tissue (Fig 6N and 6P). Finally, and as shown above (Figs 4I and 5N), YFP signals were mainly restricted to the veins in the top half of 4.5-DAG E2331>>erGFP/YFP leaves and were detected throughout the bottom half of the leaves (Fig 6O). By contrast, YFP signals were mainly restricted to the vascular tissue in the bottom half of 6.5-DAG E2331>>erGFP/YFP;*gn-13* leaves and were detected throughout the top half of the leaves—though YFP signals were still weaker in nonvascular tissues than in the vascular tissue (Fig 6Q). We conclude that *GN* controls PD-aperture-dependent vein patterning.

Vein pattern defects of intermediate alleles of *gn* are phenocopied by growth of WT in the presence of both NPA and PBA [5] (Fig 6I). Because *GN* controls PD aperture-dependent vein patterning besides auxin-transport- and auxin-signaling-dependent vein patterning, we asked whether defects in PD aperture regulation shifted the defects induced by NPA and PBA toward more severe classes of the *gn* vein patterning phenotype. To address this question, we grew E2331, E2331;*cals3-2d*, and E2331;*gsl8-2* in the presence or absence of NPA and PBA. We then imaged erGFP-expression-labeled vein networks 5 DAG in E2331 and—because *cals3-2d* and

*gsl8* leaves develop more slowly than WT leaves (Fig 4M–4O, 4Q, 4S–4X, 4Z and 4AC)—6 and 8 DAG in E2331;*cals3-2d* and E2331;*gsl8-2*, respectively.

Growth of *cals3-2d* in the presence of both NPA and PBA led to vein pattern defects similar to those of the strong *gn-van7* allele [5,12] (Fig 6J and 6K), and growth of *gsl8-2* in the presence of both NPA and PBA even phenocopied vein patterning defects of the null *gn-13* allele (Fig 6L and 6M).

We conclude that vein patterning is controlled by the *GN*-dependent, coordinated action of auxin signaling, polar auxin transport, and movement of an auxin signal through PDs.

## Discussion

Unlike the tissue networks of animals, the vein networks of plant leaves form in the absence of cell migration and direct cell–cell interaction. Therefore, leaf vein networks are patterned by a mechanism unrelated to that which patterns animal tissue networks. Here we show that leaf veins are patterned by the coordinated action of three *GN*-dependent pathways: auxin signaling, polar auxin transport, and movement of an auxin signal through PDs (Fig 7F).

### Regulation of PD permeability during leaf development

At early stages of leaf tissue development—stages at which veins are forming—PD permeability is high throughout the leaf (Fig 7A). As leaf tissues develop, PD permeability between veins and surrounding nonvascular tissues becomes gradually lower but remains high between vein cells. These results suggest that, at early stages of leaf tissue development, all cells are symplastically connected. As veins develop, vein cells remain symplastically connected but become isolated from the surrounding nonvascular tissues.

The changes in PD permeability that occur during leaf development resemble those observed during the development of embryos [40,41,69,70], lateral roots [71,72], and stomata [73–75]. By contrast, the changes in PD permeability that occur during leaf development are unlikely to be related to those observed during the transition of leaf tissues from sink to source of photosynthates as this transition begins when new veins are no longer forming and all existing veins have completely differentiated (for example, [39,42,76–78]).

Consistent with the observation that vein formation is associated first with high and then with low PD permeability between veins and surrounding nonvascular tissues, defects in PD aperture regulation—whether leading to near-constitutively wide or narrow PD aperture—lead to similar vein patterning defects: Fewer veins form, and those that do form become disconnected and discontinuous. In the most extreme cases, randomly oriented vascular elements differentiate in clusters, a phenotype that so far had only been observed in *gn* mutants or in plants impaired in both auxin signaling and polar auxin transport [5,10–12].

How symplastic connection between vein cells and their isolation from surrounding nonvascular tissues is brought about during vein development remains to be understood. One possibility is that, as in cells of the embryonic hypocotyl [41], there are more PDs along the transverse walls of vein cells than along their longitudinal walls—perhaps because no new PDs form in the longitudinal walls of vein cells during their elongation, as it happens in elongating root cells [79,80]. One other possibility is that, as it happens to elongating root cells [81], during vein cell elongation, simple PDs coalesce into branched PDs, which have narrower aperture [42]. Consistent with this possibility, there are more branched PDs along the longitudinal walls than along the transverse walls of epidermal cells underlying the midvein [82]. Yet another possibility is that the aperture of PDs along the longitudinal walls of vein cells is narrower than that of PDs along their transverse walls—perhaps because the aperture of PDs along longitudinal walls is closed by the same turgor pressure that drives cell elongation [83–

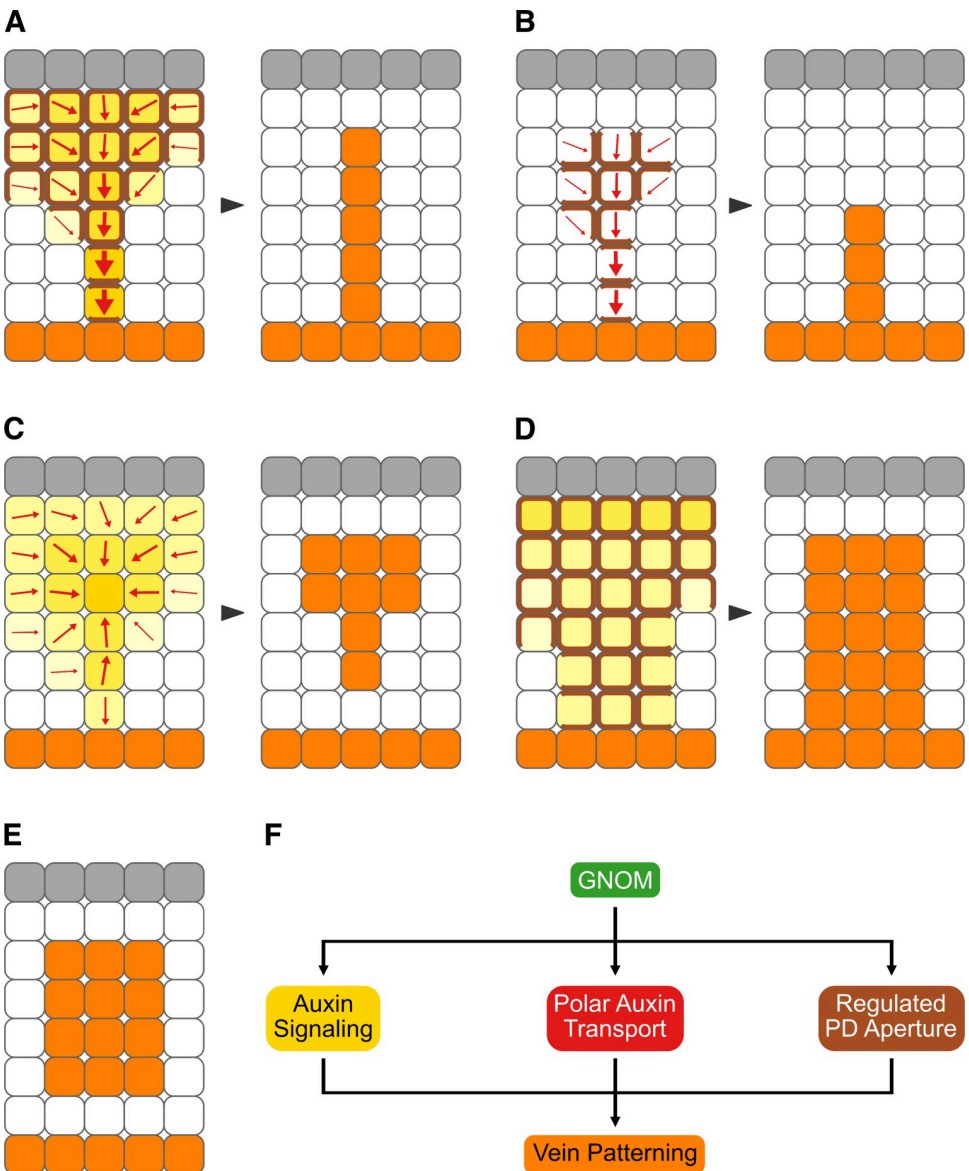

**Fig 7. Summary and interpretation.** (A–F) Gray: epidermis, whose role in vein patterning—if any—remains unclear [147]. Increasingly darker yellow: progressively stronger auxin signaling. Increasingly thicker arrows: progressively more polarized auxin transport. Brown: PD-mediated cell–cell connection. Orange: veins. Arrowheads temporally connect vein patterning stages with mature vein patterns. (A) In WT, veins are patterned by gradual restriction of auxin signaling domains [65,67,88,125,126,148], gradual restriction of auxin transport domains and polarization of auxin transport paths [5,37,38,56,65–67,126,148], and gradual reduction of PD permeability between incipient veins and surrounding nonvascular tissues. (B) Inhibition of auxin signaling leads to narrower domains of auxin transport [5,65,148] and promotes reduction of PD permeability between incipient veins and surrounding nonvascular tissues. (C) Defects in the ability to regulate PD aperture lead to weaker and broader domains of auxin signaling, fragmentation of auxin transport domains, and abnormal polarization of auxin transport paths. (D) Inhibition of auxin transport leads to weaker and broader domains of auxin signaling [5,38,48,148] and delays reduction of PD permeability between incipient veins and surrounding nonvascular tissues. (E) Loss of *GN* function or simultaneous inhibition of auxin signaling, polar auxin transport, and ability to regulate PD aperture leads to clusters of vascular cells. (F) Veins are patterned by the coordinated activities of three *GN*-dependent pathways: auxin signaling, polar auxin transport, and regulated PD aperture. *GN*, *GNOM*; PD, plasmodesma; WT, wild-type.

85] or because more callose accumulates at PDs along the longitudinal walls than at PDs along the transverse walls, as it happens in epidermal cells underlying the midvein [82]. In the future, it will be interesting to distinguish between these possibilities; however, the mechanism by which changes in PD permeability are brought about during leaf development is inconsequential to the conclusions we derive from such changes.

## Auxin, regulated PD aperture, and vein patterning

Our results suggest that auxin controls PD permeability and that regulated PD aperture controls auxin-induced vein formation. Auxin application delays the reduction in PD permeability between veins and surrounding nonvascular tissues that occurs during normal leaf development. And in the most severe cases, impaired ability to regulate PD aperture almost entirely prevents auxin-induced vein formation.

Our results suggest that also auxin signaling and regulated PD aperture control each other during vein patterning. Auxin signaling inhibition prematurely reduces PD permeability between veins and surrounding nonvascular tissues (Fig 7B), suggesting that auxin signaling normally delays such reduction. In turn, defects in the ability to regulate PD aperture lead to defects in expression of auxin response reporters (Fig 7C). Near-constitutively narrow PD aperture leads to lower levels and broader domains of expression of auxin response reporters, suggesting that an auxin signal is produced at low levels in all cells and reaches veins through PDs. Also near-constitutively wide PD aperture leads to lower levels and broader domains of expression of auxin response reporters, suggesting that high levels of an auxin signal are maintained at sites of vein formation by reducing its leakage through PDs toward surrounding nonvascular tissues.

Our findings are consistent with the inability of plants with impaired ability to regulate PD aperture to restrict expression domains and maintain high expression levels of auxin response reporters in hypocotyl and root [24,86,87]. Our interpretation is consistent with high levels of auxin signaling at early stages of vein formation and low levels of auxin signaling at late stages of vein formation [48,88]. And mutual control of auxin signaling and PD aperture regulation is consistent with the finding that simultaneous inhibition of auxin signaling and of the ability to regulate PD aperture leads to vein patterning defects that are more severe than the addition of the defects induced by auxin signaling inhibition and those induced by impaired ability to regulate PD aperture. In the most severe cases, simultaneous inhibition of auxin signaling and of the ability to regulate PD aperture leads to vascular systems comprised of very few, scattered vascular clusters.

It is unclear how auxin and its signaling could delay the reduction in PD permeability between veins and surrounding nonvascular tissues that occurs during normal leaf development. One possibility is that such delay is brought about by the ability of auxin to rapidly induce the expression of PD beta glucanases [71,89], which degrade callose at PDs and thus prevent callose-mediated restriction of PD aperture [71,90–92]. Another possibility is that the delay derives from the induction by auxin of pectin methylesterase activity [93], which localizes around PDs [94] and can increase their permeability [95–98]. A further possibility rests on the ability of auxin to reduce levels of reactive oxygen species in plastids [99], which leads to increased PD permeability [100,101]. In the future, it will be interesting to test these possibilities; nevertheless, the mechanism by which auxin and its signaling delay the reduction in PD permeability between veins and surrounding nonvascular tissues has no bearing on our interpretation of such delay.

Our results also suggest that polar auxin transport and regulated PD aperture control each other during vein patterning. Auxin transport inhibition delays the reduction in PD

permeability that occurs between veins and surrounding nonvascular tissues during normal leaf development (Fig 7D), suggesting that polar auxin transport normally promotes such reduction. In turn, defects in the ability to regulate PD aperture lead to defects in expression and polar localization of the PIN1 auxin exporter (Fig 7C), whose function is nonredundantly required for vein patterning [37]. Impaired ability to regulate PD aperture leads first to hetero-geneous PIN1 expression along continuous expression domains and then to disconnection of those expression domains from preexisting veins and breakdown of expression domains into domain fragments, suggesting that connection and continuity of PIN1 expression domains depend on regulated PD aperture. These defects in PIN1 expression resemble those of mutants in pathways that counteract *GN* function [56,102–104]. And as in those mutants, in mutants impaired in PD aperture regulation PIN1 polarity is directed away from the edge of vein frag-ments and vascular clusters. That defects in the ability to regulate PD aperture lead to defects in PIN1 expression and polar localization is consistent with reduced polar auxin transport and defective PIN2 expression and polar localization in mutants and transgenics that are impaired in PD aperture regulation [82,105].

How polar auxin transport and regulated PD aperture could control each other during vein patterning remains to be explored, but PDs are associated with receptor-like proteins [106–108] and PIN proteins with leucin-rich repeat receptor kinases [109,110], suggesting possibili-ties for the two pathways to interact. The molecular details of such interaction will have to be addressed in future research; however, our conclusion that polar auxin transport promotes the reduction in PD permeability that occurs between veins and surrounding nonvascular tissues is consistent with lower expression levels of positive regulators of callose production in auxin-transport-inhibited lateral roots [72]. Moreover, mutual control of polar auxin transport and PD aperture regulation is consistent with the finding that simultaneous inhibition of auxin transport and the ability to regulate PD aperture leads to vein patterning defects that are more severe than the addition of the defects induced by auxin transport inhibition and those induced by impaired ability to regulate PD aperture. In the most severe cases, simultaneous inhibition of auxin transport and of the ability to regulate PD aperture leads to a vasculariza-tion zone that spans almost the entire width of the leaf. However, in those leaves veins still form oriented along the longitudinal axis of the leaf, suggesting the presence of residual vein patterning activity. That such residual vein patterning activity is provided by auxin signaling is suggested by the finding that the vascular system of leaves in which auxin signaling, polar auxin transport, and the ability to regulate PD aperture are simultaneously inhibited is no more than a shapeless cluster of vascular cells.

All these observations suggest that during normal leaf development, auxin, through its sig-nal transduction, induces high PD permeability and that absence of such induction, through auxin removal by polar transport, allows PD permeability to lower between veins and nonvas-cular tissues. This conclusion seems to be inconsistent with the observation that auxin pro-motes low PD permeability during development of lateral roots and bending of mature hypocotyls [24,111] or that auxin has no effect on PD permeability in mature leaves and root tips [82,112]. However, such seeming inconsistency may simply reflect organ (leaf versus root and hypocotyl) or developmental stage (mature versus developing) responses. Indeed, auxin application induces vein formation only in developing leaves and fails to do so in mature leaves or in hypocotyls and roots of Arabidopsis [56,113].

## Control of PD aperture regulation by *GN*

The vein pattern of leaves both lacking *GN* function and impaired in the ability to regulate PD aperture is no different from that of *gn* mutants. This suggests that *GN* controls PD aperture

regulation, just as it controls auxin signaling and polar auxin transport [5]. That *GN* controls PD aperture regulation is supported by the defects in regulation of PD permeability we observed in *gn* mutants. How *GN* controls PD aperture regulation is unclear, but the most parsimonious account is that *GN* controls the localization of proteins that regulate PD aperture. This hypothesis remains to be tested but is consistent with abnormal callose accumulation upon genetic or chemical inhibition of *GN* [114].

Irrespective of how *GN* precisely controls PD aperture regulation, simultaneous inhibition of auxin signaling, polar auxin transport, and the ability to regulate PD aperture phenocopies even the most severe vein patterning defects of *gn* mutants (Fig 7E). Because vein patterning is prevented in both the strongest *gn* mutants and in the most severe instances of inhibition of auxin signaling, polar auxin transport, and the ability to regulate PD aperture, we conclude that vein patterning result from the coordinated action of three *GN*-dependent pathways: auxin signaling, polar auxin transport, and regulated PD aperture (Fig 7F).

## A diffusion-transport-based vein-patterning mechanism

The Canalization Hypothesis was proposed over 50 years ago to account for the inductive effects of auxin on vein formation [115,116]. In its most recent formulation [117], the hypothesis proposes positive feedback between cellular auxin efflux mediated by exporters polarly localized to a plasma membrane segment and polar localization of those auxin exporters to that membrane segment. The Canalization Hypothesis is supported by overwhelming experimental evidence and computational simulations; nevertheless, both experiments and simulations have brought to light inconsistencies between hypothesis and evidence (recently reviewed in [4,118]). For example, the hypothesis assumes that at early stages of auxin-induced vein formation auxin diffuses from auxin sources (for example, the applied auxin) toward auxin sinks (i.e., the preexisting veins) [116], but auxin diffusion out of the cell is unfavored over diffusion into the cell by almost two orders of magnitude [119]. Furthermore, the hypothesis assumes that the veins whose formation is induced by auxin readily connect to preexisting veins (i.e., auxin sinks)—an assumption that is supported by experimental evidence [120] but that simulations have been unable to reproduce without the addition of ad hoc solutions [66,121] or the existence of multiple auxin exporters with distinct patterns of auxin-responsive expression and polarization [122]. Finally, the hypothesis relies on the function of auxin exporters [123]—a requirement that is unsupported by experimental evidence because genetic or chemical inhibition of all the *PIN* genes with vein patterning function fails to prevent patterning of vascular cells into veins [5,34,35].

Our results suggest that those discrepancies between experiments and simulations, on the one hand, and the Canalization Hypothesis, on the other, could be resolved by supplementing the positive feedback between auxin and its polar transport postulated by the hypothesis with movement of an auxin signal through PDs according to its concentration gradient (Fig 7A). At early stages of vein formation, movement through PDs would dominate; at later stages, polar transport would take over. Computational simulations suggest that our conclusion is justified [4].

A vein patterning mechanism that combines the positive feedback between auxin and its polar transport postulated by the Canalization Hypothesis with diffusion of an auxin signal through PDs requires at least two conditions to be met. First, auxin must promote the movement of an auxin signal through PDs such that gradual reduction in PD permeability between veins and surrounding nonvascular tissues as well as maintenance of symplastic connection between vein cells are accounted for by feedback between movement of the auxin signal through PDs and PD permeability. Our results support such requirement: Auxin application

delays the reduction in PD permeability that occurs during normal leaf development, thereby promoting movement of an auxin signal through PDs. Second, auxin signaling, polar auxin transport, and movement of an auxin signal through PDs must be coupled. Were they not—for example, if PD permeability between developing veins and surrounding nonvascular tissues remained high—the high levels of auxin signaling in early stage PIN1 expression domains [67,124], which inefficiently transport auxin because of PIN1 isotropic localization [5,37,38,56,62–67], would be dissipated by lateral diffusion of the auxin signal through PDs. And if, conversely, PD permeability in tissues where veins are forming was already low, the auxin signal would not be able to diffuse toward preexisting veins, which transport auxin efficiently because of PIN1 polar localization and have low levels of auxin signaling [5,37,38,48,56,62–67]. That auxin signaling and polar auxin transport control each other during vein patterning is known [5,38,48,65,125,126] (Fig 7B and 7D). Our results support the additional requirement that polar auxin transport and movement of an auxin signal through PDs control each other and that movement of an auxin signal through PDs and auxin signaling control each other (Fig 7B–7D).

In this study, we derived patterns of PD permeability change during leaf development from movement of a soluble YFP through leaf tissues. We note that auxin, being smaller than YFP, could, for example, move from the veins to the surrounding nonvascular tissues when YFP no longer can. Nevertheless, the reduced permeability of the PDs along the longitudinal walls of vein cells suggests that less auxin moves laterally than longitudinally. Moreover, unlike YFP, auxin is the substrate of PIN exporters [6,8]. By the time YFP can no longer move out of the veins, PIN1 has become polarly localized to the basal plasma membrane of vein cells [5,37,38,56,62–67]. Such polarization drives removal of auxin—but not YFP—from vein cells [7], thereby dissipating the gradient in auxin signaling between veins and surrounding nonvascular tissues [5,38,48,56,65,67,125,126], and as such the driving force for auxin movement from the veins to the surrounding nonvascular tissues. These considerations notwithstanding, the most stringent piece of evidence in support of our conclusions would be provided by the direct visualization of auxin movement. Despite considerable advances in the visualization of auxin signals [127–129], direct visualization of auxin movement remains to this day impossible. Should this change, it would also be possible to test whether it is auxin itself or an auxin-dependent signal that moves through PDs; nevertheless, the logic of the mechanism we report is unaffected by such limitation.

Our observations pertain to vein patterning, but they may be relevant for other processes too—for example, stoma patterning. Indeed, like vein patterning, stoma patterning depends on auxin signaling [130–132], polar auxin transport [130], regulated PD aperture [25,133,134], and *GN* [11,130,135]. And as in vein patterning, stoma patterning defects in plants lacking *GN* function are quantitatively stronger than and qualitatively different from those in plants impaired in auxin signaling, polar auxin transport, or the ability to regulate PD aperture [25,130–135]. It will be interesting to understand whether the pathway network that patterns veins also patterns other plant cells and tissues.

Despite plants and animals gained multicellularity independently of each other, a mechanism similar to that which patterns plant veins and depends on the movement of an auxin signal through PDs also patterns animal tissues. At early stages of tissue development in animal embryos, cells are connected by open gap junctions such that the tissue is a syncytium (reviewed in [136,137]). And at later stages of tissue development, tissue compartments become isolated by selective closure of gap junctions to prevent unrestricted movement of signaling molecules. However, whereas in plants regulated PD aperture interacts with the polar transport of auxin and its signal transduction, in animals gap junction gating interacts with the polar secretion of signaling molecules or the binding of polarly localized ligands and receptors

protruding from the plasma membranes (reviewed in [138]). Therefore, control of vein patterning by *GN*-dependent auxin signaling, polar auxin transport, and regulated PD aperture is an unprecedented mechanism of tissue network formation in multicellular organisms.[139]

## Materials and methods

### Notation

Fusion of promoter *A* with gene *B* is indicated by double colon (i.e., A::B), in-frame fusion of gene *A* with gene *B* by single colon (i.e., A:B), and transactivation of gene *B* by enhancer A by double greater-than sign (i.e., A>>B). Unlinked mutations and transgenes are separated by semicolon.

### Plants

Origin and nature of lines, genotyping strategies, and oligonucleotide sequences are in S2–S4 Tables, respectively. Seeds were sterilized and sowed as in [140]. Stratified seeds were germinated and seedlings were grown at 23˚C under continuous light (approximately 100 µmol m$^{-2}$ s$^{-1}$). Plants were grown at 24˚C under fluorescent light (approximately 100 µmol m$^{-2}$ s$^{-1}$) in a 16-h light/8-h dark cycle. Plants were transformed and representative lines were selected as in [140].

### Chemicals

NPA and PBA were dissolved in dimethyl sulfoxide and stored at −20˚C indefinitely (NPA) or up to a week (PBA). Dissolved chemicals were added (25 µM final NPA concentration; 10 or 50 µM final PBA concentration) to growth medium just before sowing. IAA was dissolved in melted (55˚C) lanolin (1% final IAA concentration) and stored at 4˚C up to a week. Controls were treated with the sole chemical solvents.

### Imaging

Developing leaves were mounted and imaged by confocal laser scanning microscopy as in [37,57]. For each ET driver, acquisition parameters (i.e., laser transmission, detector gain, and detector offset) were first adjusted for the oldest ET>>erGFP/YFP leaves such that signals were saturated only in up to approximately 1% of the pixels in the acquired images. The same parameters were then used for younger leaves, which led to images in which signals were saturated in >1% of the pixels in the acquired images but ensured that for each ET driver all the images of ET>>erGFP/YFP leaves could be compared to one another. Mature leaves were fixed in 6:1 ethanol:acetic acid, rehydrated in 70% ethanol and water, cleared briefly (few seconds to few minutes)—when necessary—in 0.4 M sodium hydroxide, washed in water, and either (i) mounted in 1:2:8 water:glycerol:chloral hydrate and imaged by differential interference contrast or dark field illumination microscopy as in [141] or (ii) stained for 6 to 16 h in 0.2% basic fuchsin in ClearSee [142], washed in ClearSee for 30 min, incubated in daily changed ClearSee for three days, and mounted in ClearSee for imaging by confocal laser scanning microscopy. Light paths for confocal laser scanning microscopy are in S5 Table. In the Fiji distribution [143] of ImageJ [144–146], grayscaled RGB color images were turned into 8-bit images; when necessary, 8-bit images were combined into stacks, and stacks were projected at maximum intensity; look-up tables were applied to images; and image brightness and contrast were adjusted by linear stretching of the histogram.

## Supporting information

**S1 Table. Reproducibility of expression and pattern features.**
(DOCX)

**S2 Table. Origin and nature of lines.**
(DOCX)

**S3 Table. Genotyping strategies.**
(DOCX)

**S4 Table. Oligonucleotide sequences.**
(DOCX)

**S5 Table. Confocal light paths.**
(DOCX)

**S1 Fig. Phenotype classes of mature vein patterns.** Dark field illumination of mature first leaves illustrating phenotype classes (top right). Class a1: open vein network outline (A); class a3: vein fragments and/or vascular clusters (B); class a4: lobed leaf and vein fragments, and/or vascular clusters (C). Arrowheads: open loops; asterisks: vein fragments and vascular clusters. Bars: 1 mm.
(TIFF)

**S1 Data. Distribution in phenotype classes and statistical analysis of leaves in Fig 1.**
(XLSX)

**S2 Data. Proportion of length of control and PBA-grown leaves where E2331-driven YFP signal is below 75% of its maximum value.**
(XLSX)

**S3 Data. Distribution in phenotype classes and statistical analysis of leaves in Fig 6.**
(XLSX)

## Acknowledgments

We thank the Arabidopsis Biological Resource Center for *gsl8-6*/SAIL_679_H10, *gsl8-1*/SALK_111094, and *gsl8-2*/GK_851C04; Eva Benková and Jiří Friml for PIN1::PIN1:GFP; Jian Xu and Ben Scheres for PIN1::PIN1:YFP; Keiko Torii for *gsl8-chor*; Marcus Heisler and Elliot Meyerowitz for DR5rev::nYFP[HS]; Nico De Storme and Danny Geelen for *gsl8-et2*; and Yka Helariutta for *cals3-2d* and *casl3-3d*. We thank Przemek Prusinekiwcz, Mik Cieslak, and Adam Runions for insightful discussions.

## Author Contributions

**Conceptualization:** Nguyen Manh Linh, Enrico Scarpella.

**Formal analysis:** Nguyen Manh Linh, Enrico Scarpella.

**Funding acquisition:** Enrico Scarpella.

**Investigation:** Nguyen Manh Linh, Enrico Scarpella.

**Methodology:** Nguyen Manh Linh, Enrico Scarpella.

**Project administration:** Enrico Scarpella.

**Resources:** Enrico Scarpella.

**Supervision:** Enrico Scarpella.

**Validation:** Nguyen Manh Linh, Enrico Scarpella.

**Visualization:** Nguyen Manh Linh, Enrico Scarpella.

**Writing – original draft:** Nguyen Manh Linh, Enrico Scarpella.

**Writing – review & editing:** Nguyen Manh Linh, Enrico Scarpella.

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
