## [Editor Report · Decision Letter 0]

19 Jan 2022

Dear Dr Scarpella, 

Thank you for submitting your manuscript entitled "Control of Leaf Vein Patterning by Regulated Plasmodesma Aperture" for consideration as a Research Article by PLOS Biology.

Your manuscript has now been evaluated by the PLOS Biology editorial staff as well as by an academic editor with relevant expertise and I am writing to let you know that we would like to send your submission out for external peer review.

Once your full submission is complete, your paper will undergo a series of checks in preparation for peer review. Once your manuscript has passed the checks it will be sent out for review. To provide the metadata for your submission, please Login to Editorial Manager (https://www.editorialmanager.com/pbiology) within two working days, i.e. by Jan 21 2022 11:59PM.

If your manuscript has been previously reviewed at another journal, PLOS Biology is willing to work with those reviews in order to avoid re-starting the process. Submission of the previous reviews is entirely optional and our ability to use them effectively will depend on the willingness of the previous journal to confirm the content of the reports and share the reviewer identities. Please note that we reserve the right to invite additional reviewers if we consider that additional/independent reviewers are needed, although we aim to avoid this as far as possible. In our experience, working with previous reviews does save time. 

If you would like to send previous reviewer reports to us, please email me at ialvarez-garcia@plos.org to let me know, including the name of the previous journal and the manuscript ID the study was given, as well as attaching a point-by-point response to reviewers that details how you have or plan to address the reviewers' concerns. 

Given the disruptions resulting from the ongoing COVID-19 pandemic, please expect some delays in the editorial process. We apologise in advance for any inconvenience caused and will do our best to minimize impact as far as possible.

Kind regards,

Ines

--

Ines Alvarez-Garcia, PhD

Senior Editor

PLOS Biology

---

## [Decision Letter · Decision Letter 1]

26 Feb 2022

Dear Dr Scarpella,

Thank you for submitting your manuscript entitled "Control of Leaf Vein Patterning by Regulated Plasmodesma Aperture" for consideration as a Research Article at PLOS Biology. Thank you also for your patience as we completed our editorial process, and please accept my apologies for the delay in providing you with our decision. Your manuscript has been evaluated by the PLOS Biology editors, an Academic Editor with relevant expertise, and by four independent reviewers.

As you will see, the reviewers find the conclusions novel and interesting, but they also ask for several clarifications and a few experiments to strengthen the findings. Both Reviewers 2 and 3 raise concerns regarding the lack of direct evidence demonstrating that GNOM controls permeability of plasmodesmata and suggest some experiments to address it. After discussing the reviews with the Academic Editor, however, we have decided not to make this a requirement for publication.

In light of the reviews (attached below), we will not be able to accept the current version of the manuscript, but we would welcome re-submission of a revised version that takes into account the reviewers' comments. We cannot make any decision about publication until we have seen the revised manuscript and your response to the reviewers' comments. Your revised manuscript is also likely to be sent for further evaluation by the reviewers.

We expect to receive your revised manuscript within 3 months. 

**IMPORTANT - SUBMITTING YOUR REVISION**

3. Resubmission Checklist

a) *PLOS Data Policy*

b) *Published Peer Review*

d) *Blurb*

Please also provide a blurb which (if accepted) will be included in our weekly and monthly Electronic Table of Contents, sent out to readers of PLOS Biology, and may be used to promote your article in social media. The blurb should be about 30-40 words long and is subject to editorial changes. It should, without exaggeration, entice people to read your manuscript. It should not be redundant with the title and should not contain acronyms or abbreviations. For examples, view our author guidelines: https://journals.plos.org/plosbiology/s/revising-your-manuscript#loc-blurb

Sincerely,

Ines

--

Ines Alvarez-Garcia, PhD

Senior Editor

PLOS Biology

Reviewers' comments

Rev. 1:

The authors investigate the role of plasmodesmatal aperture regulation on leaf venation patterning in Arabidopsis. By using an array of narrow and wide plasmodesmata aperture mutants, combined with a series of ET>>erGFP/YFP enhancer trap drivers, the authors show that plasmodesmatal permeability changes during vein development. Defects in plasmodesmatal aperture influence auxin transport and signalling. Plasmodesmatal aperture depends on the action of the regulator GNOM (GN) and simultaneous inhibition of PD aperture, auxin transport and auxin signalling phenocopies gn mutants. Vein patterning takes place through three GNOM-dependent pathways controlling auxin transport, signalling and plasmodesmata aperture.

General considerations:

The combined use of the ET>>erGFP/YFP constructs with wide and narrow aperture mutants is a very clever system for this type of investigation. The results are intriguing and shed light on the role of plasmodesmatal permeability on vein patterning. The model reported in Fig.7 is effective in clarifying the findings.

Although I appreciate the thoroughness of the manuscript, it is too long. It would be more effective if shorter and more concise, with better outlined methods.

Detailed comments on the manuscript.

Abstract and introduction:

Please consider a more careful choice of wording for the last sentence in both your abstract and introduction sections. Vein patterning may proceed with different mechanisms in other species.

Vein classification terminology:

The terminology used to indicate vein ranks is inconsistent. For example, Fig.1a shows a nice schematic of the terminology used to indicate Arabidopsis vein ranks (midvein, loops and minor veins). However, dotted around in the main text definitions such as "laterals" appear. Please clarify.

Materials and methods:

Please state what the corresponding controls were in the "chemicals" section.

Very little information is provided regarding the parameters used for the subdivision of phenotypes into classes. It would be useful to have a general description of how classes were sorted.

Regarding the "imaging" section: were the same parameters used for imaging of YFP in all erGFP/YFP samples and thus, are the different panels comparable? Some images seem a bit overexposed, particularly those of young leaves. Was this was done on purpose to compare the samples or should different panels be considered separately instead.

Results:

Fig.1:

The classification of phenotypes is quite confusing. For example, classes 0, a2 and a6 in Fig.1 are evidently different; however, I struggle to see the difference between a2 and a5 and I don't know what a1, a3 and a4 look like since there is no representative image in the main text, nor within the supplementals. Adding some more information about what the different phenotype classes look like, and a corresponding methodology section in the Materials & Methods will help clarify this point.

Fig.3:

Please indicate the site of IAA application.

Fig.4:

I couldn't help but notice that the gsl8-chor line was used for panels Y-AC, instead of the gsl8-2 line used throughout the rest of the manuscript. Quite likely they behave similarly, given the classification shown in Fig.1 and the fact that both are strong mutants. Did you also test the gsl8-2;PIN1-YFP line and, if not, what was the motivation for choosing the gsl8-chor line instead.

Fig.5:

What is the difference between the DR5rev-nYFPHS and DR5rev-nYFPES reporters? At first glance it seems that the same DR5rev repeats were used for both constructs. I am puzzled by the fact that the same stage of development (4DAG) displays different DR5 patterns (panels 5P and 5R): as an example, there is no strong DR5 activation at the leaf tip in 5R, compared to 5P. Furthermore, it is not clear to me what mechanism could lead to lower and broader DR5 activation in both the DR5rev-nYFPHS;cals3-2d and DR5rev-nYFPHS;gsl8-2 lines.

Fig. 6:

As for Fig. 1, concerns about phenotype classification stand for Fig. 6 too.

Furthermore, Fig.6g lacks quantification of classes for gn-13;gsl8-2 mutants; my understanding is that, presumably, the distribution would be similar to that of gn-13. Is there a reason why this was not quantified, since at p.17 you say that phenotypes of gn-13;cals3-3d, gn-13;cals3d-2d and gn-13;gsl8-2 are equivalent to those observed for gn-13?

Discussion:

The discussion is generally quite thorough and I appreciated the summary model in Fig.7. However, the "Control of PD aperture regulation by GN" section could be expanded, given that this is one of the key points of your work on which the model in Fig.7 is built on. Is there any evidence of GN regulating or interacting with proteins that regulate PD aperture?

On the same note: at times, various potential explanations of the results are provided but it's not clear which you consider more likely in the context of your model. That is the case for example, on pages 19 and 21. The discussion could be more focused and concise to better support your Fig.7 model.

Rev. 2:

This manuscript studies the mechanisms of auxin-induced leaf vein patterning. Through a successful combination of established mutant lines and pharmacological approaches, the authors have found the importance of the movement of auxin or an auxin-dependent signal through PDs. This could be a missing piece in the GN-dependent vein-patterning pathway, which interacts in a coordinated manner with two other pathways, auxin signalling and auxin polar transport. This has a potential to be accepted to the PLOS Biology, but there are still some concerns that needed to be addressed.

Major:

Lack of vascular tissue continuity seems to be a consequence of an early defect. At that early stage, GFP moves very freely among all leaf cells suggesting that PDs are not very tightly regulated at such early time point. It would be important to provide more direct evidence that mutant backgrounds used in the study are directly involved in leaf vascular tissue development, that could include:

- assessing expression pattern of Cals3 and GLS8 in the context of leaf vein development

- assessing levels of callose accumulation in the cells of developing vascular tissue

- assessing PD permeability in the mutant backgrounds used as tools for studying PD permeability (cals3, gls8). The authors should include those two mutant lines in the fluorescent protein mobility assay. Future approach could benefit from gauging PDs with bigger proteins like 2xYFP, 3xYFP

The study is missing direct evidence that GNOM controls permeability of PDs (e.g movement of GFP in the gn mutant; callose levels at the WT and mutant's PDs).

The authors wrote that strong vascular phenotypes like vascular clustering, may occur in the strong mutant of both pathways: polar auxin transport plus auxin signalling. Both of these pathways are controlled by GNOM. Are such mutants affected in PD permeability?

Fig. 3 shows only the images only from before and after the treatment. It's hard to assess how images showing vein loops after the treatment differ from untreated leaf. Please provide images (supplementary, time course) of intermediated stages, clearly illustrating effect of auxin application.

The difference between Figs 5N and 5O is unclear. It should be quantified in some way.

Many of the sentences are too long and thus hard to follow. Few examples are:

"To test this prediction, we applied the natural auxin indole-3-acetic acid (IAA) to one side of 3.5-DAG first leaves of E2331 and Q0990, and — because cals3-2d and gsl8 leaves develop more slowly than WT leaves (see below) — of 4.5-DAG first leaves of E2331;cals3-2d and Q0990;gsl8 (Fig. 3A,B,E,F)."

"Moreover, only in ~30% (8/27) of the E2331;cals8-2d leaves in which veins did form in response to IAA application did these veins connect to the midvein: in the remaining ~70% of the responding leaves, the veins whose formation was induced by IAA application ran parallel to the midvein through the leaf petiole (Fig. 3D)."

"Should the auxin-transporter-unmediated movement of an auxin signal that controls vein patterning be mediated by PDs, defects in PD aperture regulation would enhance vein patterning defects induced by auxin transport inhibition."

Minor:

It would be easier to follow the information in the figures if all panels were described with genotype and treatment. This could be resolved with the annotation above or below the image itself.

Rev. 3:

Scarpella and Linh focuses on understanding the molecular pathways involved in vascular development and leaf patterning. The authors have previously addressed the role of auxins in this developmental process and identified GN as a regulatory signalling hub. Here, the authors built on these findings and propose a role for plasmodesmata in this process. There are previous evidences showing Plasmodesmata contribute to auxin distribution mediated by the expression of callose synthases (e.g., GSL8 by Han et al., 2014) and modelling approaches (Mellor et al, 2020). There is also previous papers showing the role of plasmodesmata in vascular development. Based on those it is not surprising that when the authors modify plasmodesmata aperture (using GSL mutant lines), they find defects in vascular development and auxin transport. The novelty of this paper is in relating this auxin-callose regulatory pathway with the signalling hub GN which acts in leaf vascular development thus my comments below focus on the questions: is there enough evidences to support a GN-dependent pathway regulating callose at plasmodesmata and symplasmic auxin transport? To summarize my conclusions, I think the authors did a lot of work to demonstrate the link between callose synthesis, symplasmic transport, vascular development and auxin but, the link to GN in my opinion is weak and lack mechanistic understanding. There is merit in showing that auxin signalling interacts with callose regulation in the context of leaf patterning but not enough evidence to demonstrate that GN-dependent signalling is directly involved and the effect this has on plasmodesmata structure and function. The authors should moderate their conclusion and interpretation of the experiments provided here.

Major comments:

1- Abstract claims: 'Therefore, veins are patterned by the coordinated action of three GN-dependent pathways: auxin signaling, polar auxin transport, and movement of an auxin signal through PDs. We have identified all the key vein-patterning pathways in plants' How the authors know that 1) there is no evidence excluding other key-vein patterning pathways. The absence of evidence of not evidence of absence and 2) the results do not necessarily demonstrate the movement of an auxin signal through PDs, they indicate that a signal via PD contribute to auxin distribution. This is not new as per reported by Han et al., 2014 and Mellor et al., 2020 to cite 2 examples.

2- Related to the above, end of intro indicates ' In the most severe cases, the vascular system of leaves in which those three pathways have been inhibited is no more than a shapeless cluster of vascular cells, suggesting that we have identified all the main pathways in vein patterning' these mutants, inhibitor treatments have many other phenotypes, that could be interacting or even responsible of the leaf vascular patterning, How the authors can read on the leaf vascular patterning alone to get this conclusion without considering these many other defects? Casl3 and gsl8 affects whole plant development (stomata development, root growth, meristem formation, cell plate formation, cytokinesis etc.).

3- Start of the result 'Here we tested the hypothesis that the movement of an auxin signal that controls vein patterning and that is not mediated by auxin transporters is mediated by PDs.' I like this very much. This is the right framing of their hypothesis, which left me wondering why this is oversold/ overinterpreted elsewhere in the paper?

4- Section Control of Vein Patterning by Regulated PD Aperture: The authors should consider using different plasmodesmata mutants to clearly link plasmodesmata to vein formation. So far the results only demonstrate that callose regulation (these mutants are not only affecting PD-callose but also callose in cell plates and sieve pores etc. ) affects leaf vascular development, among the many other phenotypes these mutants have. Vaten et al., Dev cell 2013 already described the effect of cals3 in vascular development thus I am not sure how much here it is new.

5- Note that callose synthesis at plasmodesmata is counteracted by its degradation, thus there is a need to evaluate the expression profile and regulation of beta 1,3 glucanases in this context.

6- the use of chemicals and inhibitors to address auxin role is difficult to interpret specially in the callose mutant phenotypes. Plasmodesmata and callose are very susceptible to changes in the media, including mild osmotic changes. Has this be considered?

7- Is plasmodesmata regulation affected in gn? Expression of symplasmic reporters in this mutant background could help address this question. Maybe electron microscopy is necessary to visualize plasmodesmata in the mutants.

8- The authors conclude epistatic effects in gn,cals double mutants but gn13 mutants already have a very strong phenotype that would obscure any evaluation of epistatic or additive effects.

9- The discussion is extensive (10 pages) and could be easily condensed. The last section is unnecessary and distracting.

10- Figure legends, the structure of the legends is very confusing to understand. Title are the author's conclusions and there is no detail description of what is shown and how the experiment was performed, instead there is a list of abbreviators, colour grades and objects. I would prefer if the author take the time to describe the experiment and the results shown in each panel.

Rev. 4:

This manuscript reports how movement of an auxin signal through plasmodesmata is essential for correct vein formation within the leaf. The study presents a detailed analysis of this process, clearly describing hypotheses and carefully characterising the vein patterns in numerous relevant mutants. Given vein formation is a popular research area, with a long history, the idea that this could be a key refinement of the 'Canalisation hypothesis' is an exciting prospect. Furthermore, auxin diffusion through plasmodesmata has received much interest recently, with several highly cited papers showing this to be an important process in other organs in the last few years. I would therefore expect this manuscript to be of wide and long-standing interest.

Clearly a huge amount of experimental work has gone into the manuscript, however, I felt in places the relevance of this work in the field, and mechanistic insights could be improved. The work clearly is a key refinement of the Canalisation hypothesis, and yet discussions of the coupling between auxin concentration, and auxin-regulation of PIN transport and plasmodesmatal permeability was delayed until late in the discussion. I have a number of suggestions that I feel would improve this, and generally improve the readability of the manuscript:

1. In the introduction, I was left wondering (i) whether it was previously known that plasmodesmata play a role in vein formation, and (ii) whether there was previous evidence that plasmodesmata are regulated by GNOM. You write "here we ask whether movement of auxin or an auxin-dependent signal through PDs is one of the missing GN-dependent

vein-patterning pathways", but I feel more details are needed as to what was known about this before. It is written of p8 "Consistent with previous observations (Kim et al.)," however, details of Kim et al are not in the introduction. Were these previous observations part of your motivation for suggesting your hypothesis? It would be good to clarify at the outset what was already known.

2. The introduction also briefly mentions that auxin signalling has been shown to play a role in vein formation, however, given these findings are built on in this manuscript, again I felt more details are needed. On p15, you write "auxin-signaling-unmediated movement of an auxin signal" which seems to suggest that it is thought that the auxin signalling is acting through auxin movement. I suggest clarifying this in the introduction.

3. On p5, you write "As calsP-d, gslR mutants formed networks" Please rephrase, as at first I thought you were talking about the double mutant.

4. On p9, you write "To test this prediction, we applied the natural auxin indole-S-acetic acid (IAA) to one side of 3.5-DAG first leaves of E2331" and Q0990" Presumably these are the control lines for the subsequently stated mutants - could this be clearer.

5. On p11, "Should the auxin-transporter-unmediated movement of an auxin signal that controls vein patterning be mediated by PDs," It would be good to be clearer here as to what is known and what the hypothesis is - the text seems to suggest that some sort of alternative auxin movement has already been established.

6. The phrase "auxin-transporter-unmediated movement" is somewhat clumsy, and is used in several places - do you simply mean passive movement? This leads on to phrases such as "Should the residual, auxin-transporter- and auxin-signaling-unmediated movement of an auxin signal that controls vein patterning be mediated by PDs" which is somewhat hard to follow! Please rephrase.

7. I also disagree that the phrases involving "PD mediate auxin movement" is accurate - the verb mediate tends to be used for indirect influences, and so is perhaps not suitable to describe how auxin passively moves through the plasmodesmata. I would have said that rather the PD enable passive auxin movement.

8. p13 describes the PIN1 distribution, could you clarify whether PIN1 exhibits a polar localisation within the cells.

9. Discussion, p18 "we have identified all the main pathways that control vein patterning." Presumably 'we' is meaning the community, rather than the authors of this paper - may be wise to rephrase!

10. Discussion, p22, That auxin itself is known to regulate plasmodesmatal permeability is a key mechanistic insight that has major implications for this study. I suggest that this point is made earlier. One possibility is that auxin itself is controlling the plasmodesmata dynamics during normal leaf development - I felt this point could be made more explicitly. Previous studies have shown that plasmodesmatal permeability is reduced by auxin however, which seems counterintuitive given the results presented here show high plasmodesmatal permeability within the veins - perhaps this could be discussed.

11. Discussion, p22. I believe that Han et al also showed auxin-regulation of plasmodesmatal permeability, focussing on gsl8, and showed this regulation to be through the auxin signalling pathway.

12. The discussion section "A Diffusion-Transport-Based Vein-Patterning Mechanism" provided clear insights into that plasmodematal auxin diffusion together with auxin regulation of plasmodesmatal permeability could couple to the canalisation mechanism involving PINs to fully explain vein formation. However, I felt some of these ideas could be introduced early. For instance, that it's established that auxin can regulate plasmodesmatal permeability seems to be key to the results described on p9-10. That coupling with auxin-mediated PIN regulation is also thought to be important for vein formation could help understanding of the results on p11-14. Furthermore, an earlier section of the discussion "Regulation of PD Permeability During Leaf Development" doesn't mention auxin regulation, and concludes "the mechanism by which changes in PD permeability are brought about during leaf development is inconsequential to the conclusions we derive from such changes." As mentioned above, I felt that introducing these mechanistic insights earlier and more explicitly would make the paper stronger.

---

## [Decision Letter · Decision Letter 2]

22 Jul 2022

Dear Dr Scarpella,

Thank you for your patience while we considered your revised manuscript entitled "Control of Leaf Vein Patterning by Regulated Plasmodesma Aperture" for publication as a Research Article at PLOS Biology. This revised version of your manuscript has been evaluated by the PLOS Biology editors, the Academic Editor and three of the original reviewers.

Based on the reviews, we are likely to accept this manuscript for publication, provided you satisfactorily address the following data and other policy-related requests (see below).

In addition, we would like you to consider a suggestion to improve the title:

"Leaf vein patterning is regulated by the aperture of plasmodesmata intercellular channels"

We expect to receive your revised manuscript within one week. 

*Published Peer Review History*

*Press*

Sincerely,

Ines

--

Ines Alvarez-Garcia, PhD

Senior Editor

PLOS Biology

SPECIES INDICATED IN THE ABSTRACT? 

- Please note that per journal policy, the model system/species studied should be clearly stated in the abstract of your manuscript.

Reviewers' comments

Rev. 2:

We think the authors adequately answered our questions.

Rev. 3:

The authors have answered to my comments in the last version.

Rev. 4:

This is an interesting, novel and detailed study that reveals how auxin diffusion through plasmodesmata contributes to vein patterns. Given the interest in this area, I am sure this would attract a wide readership. The authors have addressed my previous concerns and I have no further suggestions.

---

## [Editor Report · Decision Letter 3]

3 Aug 2022

Dear Dr Scarpella,

Thank you for the submission of your revised Research Article entitled "Leaf Vein Patterning Is Regulated by the Aperture of Plasmodesmata Intercellular Channels" for publication in PLOS Biology. On behalf of my colleagues and the Academic Editor, Mark Estelle, I am happy to say that we can in principle accept your manuscript for publication, provided you address any remaining formatting and reporting issues. These will be detailed in an email you should receive within 2-3 business days from our colleagues in the journal operations team; no action is required from you until then. Please note that we will not be able to formally accept your manuscript and schedule it for publication until you have completed any requested changes.

PRESS

Sincerely, 

Ines

--

Ines Alvarez-Garcia, PhD

Senior Editor

PLOS Biology
